# Parallel ventral hippocampus-lateral septum pathways differentially regulate approach-avoidance conflict

Dylan C. M. Yeates[1], Dallas Leavitt[1], Sajeevan Sujanthan [1], Nisma Khan[1], Denada Alushaj[1], Andy C. H. Lee [1,2] & Rutsuko Ito [1,3✉]

The ability to resolve an approach-avoidance conflict is critical to adaptive behavior. The ventral CA3 (vCA3) and CA1 (vCA1) subfields of the ventral hippocampus (vHPC) have been shown to facilitate avoidance and approach behavior, respectively, in the face of motivational conflict, but the neural circuits by which this subfield-specific regulation is implemented is unknown. We demonstrate that two distinct pathways from these subfields to lateral septum (LS) contribute to this divergent control. In Long-Evans rats, chemogenetic inhibition of the vCA3- LS caudodorsal (cd) pathway potentiated approach towards a learned conflict-eliciting stimulus, while inhibition of the vCA1-LS rostroventral (rv) pathway potentiated approach non-specifically. Additionally, vCA3-LScd inhibited animals were less hesitant to explore food during environmental uncertainty, while the vCA1- LSrv inhibited animals took longer to initiate food exploration. These findings suggest that the vHPC influences multiple behavioral systems via differential projections to the LS, which in turn send inhibitory projections to motivational centres of the brain.

[1] Department of Psychology (Scarborough), University of Toronto, Toronto, ON M1C 1A4, Canada. [2] Rotman Research Institute, Baycrest Centre, Toronto, ON M6A 2E1, Canada. [3] Department of Cell and Systems Biology, University of Toronto, Toronto, ON M5S 3G5, Canada. ✉email: rutsuko.ito@utoronto.ca

Making effective judgements about the value of environmental stimuli, and deploying optimal behavior is an essential part of survival and everyday decision making that requires the coordinated activation of a complex set of neuropsychological processes and systems. Organisms often encounter stimuli that predict both positive and negative outcomes simultaneously, creating an approach-avoidance conflict scenario that must be resolved before an action is taken. A complete delineation of the neural subsystems underlying such approach-avoidance conflicts is yet to be achieved and remains an important goal[1,2], since disruption of these systems is thought to underlie many psychiatric conditions[3], including anxiety disorders, eating disorders, and substance abuse[4–7].

While many neural substrates, including the prefrontal cortex, striatum and paraventricular thalamus[8–10], have been found to regulate aspects of approach-avoidance conflict, the ventral hippocampus (vHPC; anterior HPC in primates) has recently emerged as a critical node in the arbitration of approach-avoidance conflict elicited by the presentation of stimuli signaling incompatible goals[11–16]. The vHPC has long been recognized as a key mediator of anxiety-like behavior in rodents, which is closely associated with brain regions that control motivated behaviors[17–23]. In approach-avoidance conflict scenarios, lesions that encompass the vHPC potentiate approach toward learned stimuli that predict both appetitive and aversive outcomes[11], result in greater time spent in bright and open places rodents naturally avoid[21], and regulate feeding in conditions of uncertainty[20,24], suggesting that under normal conditions the vHPC inhibits approach responses to motivationally conflicting stimuli. However, we recently observed that pharmacological manipulation of different vHPC subfields led to divergent effects during learned approach-avoidance conflict, with inactivation of either the ventral DG (vDG) or ventral CA3 (vCA3) potentiating approach toward conflict stimuli, and inactivation of the ventral CA1 (vCA1) leading to overall avoidance of the conflict stimulus[12,13]. These findings raise the possibility that the vDG/vCA3 and vCA1 exert independent control over approach-avoidance conflict via differential projections to downstream targets.

One such candidate downstream target is the lateral septum (LS), which is the only area in the brain that receives projections from both the vCA3 and vCA1[17,19]. These inputs are organized topographically, with the caudodorsal LS (LScd) receiving the only extra-hippocampal glutamatergic efferents originating from the vCA3, while the rostral and ventral domains of the LS (LSrv) receive vCA1 and ventral subiculum inputs[17,19,25]. These LS domains are also demarcated by domain-specific histochemical markers and relationships with parts of the hypothalamus and midbrain[26,27]. The LS has been implicated in many motivational processes, although a specific functional description for the region has been elusive, with this region being associated with behaviors as divergent as the regulation of feeding[28–30], navigation[31–33], and responses to anxiolytic or stressful conditions[34–38].

Given the established role of the LS in motivated behavior and the prominence of its vHPC inputs, we hypothesized that the glutamatergic vCA3 to LScd pathway (vCA3 → LScd) and vCA1 to LSrv pathway (vCA1 → LSrv) would play critical but differential roles in approach-avoidance conflict resolution, with the former pathway mediating avoidance and the latter facilitating approach under motivational conflict. Using a combination of chemogenetics and well-established innate and learned approach-avoidance conflict tasks[11,39] (Fig. 1), we found in accordance with our hypotheses that inhibition of the vCA3 → LScd pathway induced increased approach responses toward learned conflict stimuli and led to faster oral exploration of food in a novel, potentially dangerous environment. In contrast to our predictions, however, inactivation of the vCA1 → LSrv pathway led to a non-specific behavioral disinhibition in the presence of learned conflict stimuli, and slowed the time that the animals took to make contact with the food in a novel environment. Collectively, we have identified a vCA3 → LScd pathway that suppresses exploratory responses toward motivationally significant stimuli in favor of avoidance responses, and a vCA1 → LSrv pathway that non-specifically attenuates approach responses toward motivationally salient stimuli under situations of learned and innate approach-avoidance conflict.

## Results

**The vCA3 and vCA1 send distinct glutamatergic projections to the LS.** Long-Evans rats were transfected with either AAV8-CAMKII-hM4Di-mCherry or AAV8-CAMKII-EGFP control viruses, targeting the glutamatergic cell population of the vCA3 (Fig. 2a) or vCA1 (Fig. 2e). Viral infusions targeting the vCA3 in either hM4Di or EGFP groups consistently showed fluorescent tag expression along the vCA3 (Fig. 2b, d), while infusions targeting the vCA1 resulted in consistent transduction in the ventral CA1 (Fig. 2f, h) with some unilateral spillover to adjacent subfields, or cortical areas which do not send substantial projections to the LS regions targeted in this experiment. Fluorescent tags linked to the axonal fibers were observed in the LS. When the transduction was confined to the vCA3, projections to the LS were exclusively observed throughout the lateral part of the dorsal LS (Fig. 2b), while vCA1 specific transduction resulted in projections in more rostral and ventral parts of the LS (Fig. 2f), consistent with previous literature[17,19,40]. Bilateral cannulae were implanted to carefully chosen terminal targets in the LS (caudodorsal (cd) vs. rostralventral (rv)) to avoid unnecessary spread of clozapine-N-oxide (CNO) to overlapping/non-targeted parts of the LS (Fig. 2c, g). hM4Di animals infused with CNO prior to sacrifice showed significantly attenuated c-Fos activity compared to their EGFP controls, as well as saline treated animals (Fig. 2i–l), demonstrating that inactivation of the ventral hippocampus (vHPC) inputs decreased overall activity in the targeted part of the LS. Although higher c-Fos activity was detected in the LSrv groups compared to the LScd groups overall, there was no interaction between region and other factors.

**Mixed valence cue learning.** Rats were trained on the previously validated mixed valence Y-maze conflict task[11–13], where they learned to associate distinct visuo-tactile bar cues with either sucrose reward, mildly aversive foot shock, or no outcome (Fig. 1). After every 4th training session, the animals were allowed to freely explore all three cues in extinction. During their final cue acquisition test, animals in both the vCA3 → LScd and vCA1 → LSrv groups spent the most time exploring the arm containing the cue associated with sucrose availability, the least time in the aversive cued arm, and an intermediate amount the neutral arm, indicating successful learning of the cue-outcome associations (Supplementary Fig. 1a, e). All animals showed similar exploration patterns regardless of virus type and future drug infusion condition. Well trained animals also demonstrated several behaviors that were specific to the valence of the cues. Rats chose to enter the appetitively cued arm more than the neutral arm, and the neutral arm more than the aversively cued arm (Supplementary Fig. 1b, f). "Arm-stays", which occurred when an animal exiting an arm turned around and returned to the well as an indicator of active reward-seeking, were observed mostly in the appetitive arm, seldom in the neutral, and almost never in the aversive arm (Supplementary Fig. 1c, g). Finally, animals exhibited "retreat" behaviors regularly in response to the aversive arm, which consisted of reluctant half entries into an aversive arm

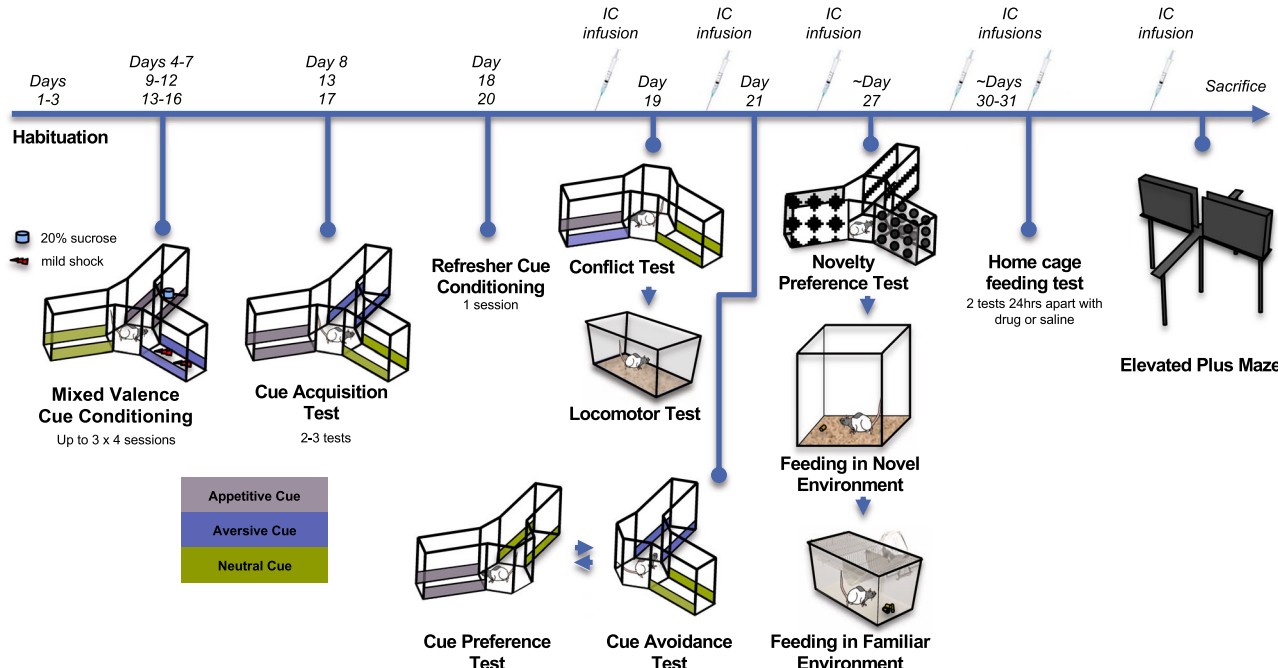

**Fig. 1 Timeline of experiments.** Following surgery, animals were trained in the mixed valence Y-maze task first. Following a successful cue acquisition test, animals received intracranial microinfusions of CNO (1 mM, depicted as IC infusion) and were administered a conflict test and locomotor test. After a refresher conditioning session, animals were infused again, and then administered the appetitive and aversive preference tests. Next, animals were given ~5 days of free feeding, and then food restricted for a novelty detection test, novel environment suppressed feeding (NESF) and the familiar environment suppressed feeding test (FESF), prior to which CNO was microinfused. Following these tests, the animals were given free access to food for 24 h, and were then food restricted again prior to the two free feeding tests, which occurred on successive days. Finally, animals received a final infusion of either CNO or saline, were administered the elevated plus maze, and sacrificed thereafter for brain extraction.

followed by withdrawal, as well as rapid back-treading when inside the arm (Supplementary Fig. 1d, h).

**Inactivation of vCA3 → LScd circuit increases conflict approach and suppresses avoidance responses, whereas inactivation of vCA1 → LSrv circuit increases non-specific approach responses.** Following successful cue-outcome acquisition, rats were infused with CNO and placed in the Y-maze with access to two arms, with one arm containing the previously conditioned neutral cues and the other a compound conflict arm containing both appetitive and aversive cues (Fig. 3a). Previous studies have shown that control rats trained on the mixed valence task display approximately equal approach/avoidance tendencies between the conflict and neutral stimuli[4,11–13,41]. Consistent with this, the EGFP vCA3 → LScd group spent equal times in both arms on average and displayed a balanced approach-avoidance ratio. However, hM4Di vCA3 → LScd rats showed a large approach bias toward the conflict arm (Fig. 3b), due to this group spending far longer exploring the conflict arm than the neutral (Fig. 3c). Animals in the hM4Di vCA3 → LScd group also made fewer entries into the neutral arm than the conflict arm, whereas the control group made approximately equal entries into both arms (Fig. 3d). It was also found that the hM4Di vCA3 → LScd rats showed a decline in the number of retreats compared to their controls, which was driven primarily by a decrease in the number of retreats in response to the conflict arm (Fig. 3e). The number of stays was higher in the conflict arm than the neutral, although this behavior was unaffected by virus type (Fig. 3f), which suggests that all animals were more likely to recognize the conflict arm as being positively valanced to some degree compared to the neutral arm. There was also no noticeable effect on the latency to enter either arm (Fig. 3g). Together, these findings suggest that inactivation of

the vCA3 → LScd circuit increased approach behavior to cues eliciting motivational conflict.

In contrast to the vCA3 → LScd inactivated animals, vCA1 → LSrv inactivated animals showed potentiated approach toward both the conflict and neutral arms, which on average resulted in equal approach-avoidance preferences that did not differ from the control group (Fig. 3h, i). The animals also showed fewer retreats from both arms compared to controls (Fig. 3k), consistent with the idea that these inactivated animals were generally disinhibited. In contrast to the vCA3 → LScd inactivated animals, the vCA1 → LSrv inactivated animals were not more likely to enter any given arm than their controls (Fig. 3j). Much like the previously described groups, both inhibited and control vCA1 → LSrv animals trended toward showing more arm stays in the conflict arm compared to the neutral (Fig. 3l) but showed no differences in entrance latency regardless of arm identity or virus type (Fig. 3m).

Overall, the vCA3 and vCA1 pathways to the LS appear to mediate overlapping but distinct behaviors. Under normal conditions, the vCA3 → LScd pathway may mediate behavioral inhibition in favor of conflict avoidance. In contrast, the vCA1 pathway may mediate behavioral inhibition in response to both motivationally conflicting and to motivationally neutral stimuli.

**vHPC to LS pathways do not regulate preferences for appetitive or aversive conditioned stimuli alone.** It is possible that the behaviors induced by circuit inhibition are a result of changes in motivation to either approach reward-predictive cues or avoid shock-predictive cues, or memory for the cue value. To assess these possibilities, we infused the rats with CNO and administered simple conditioned cue preference tests, allowing them to

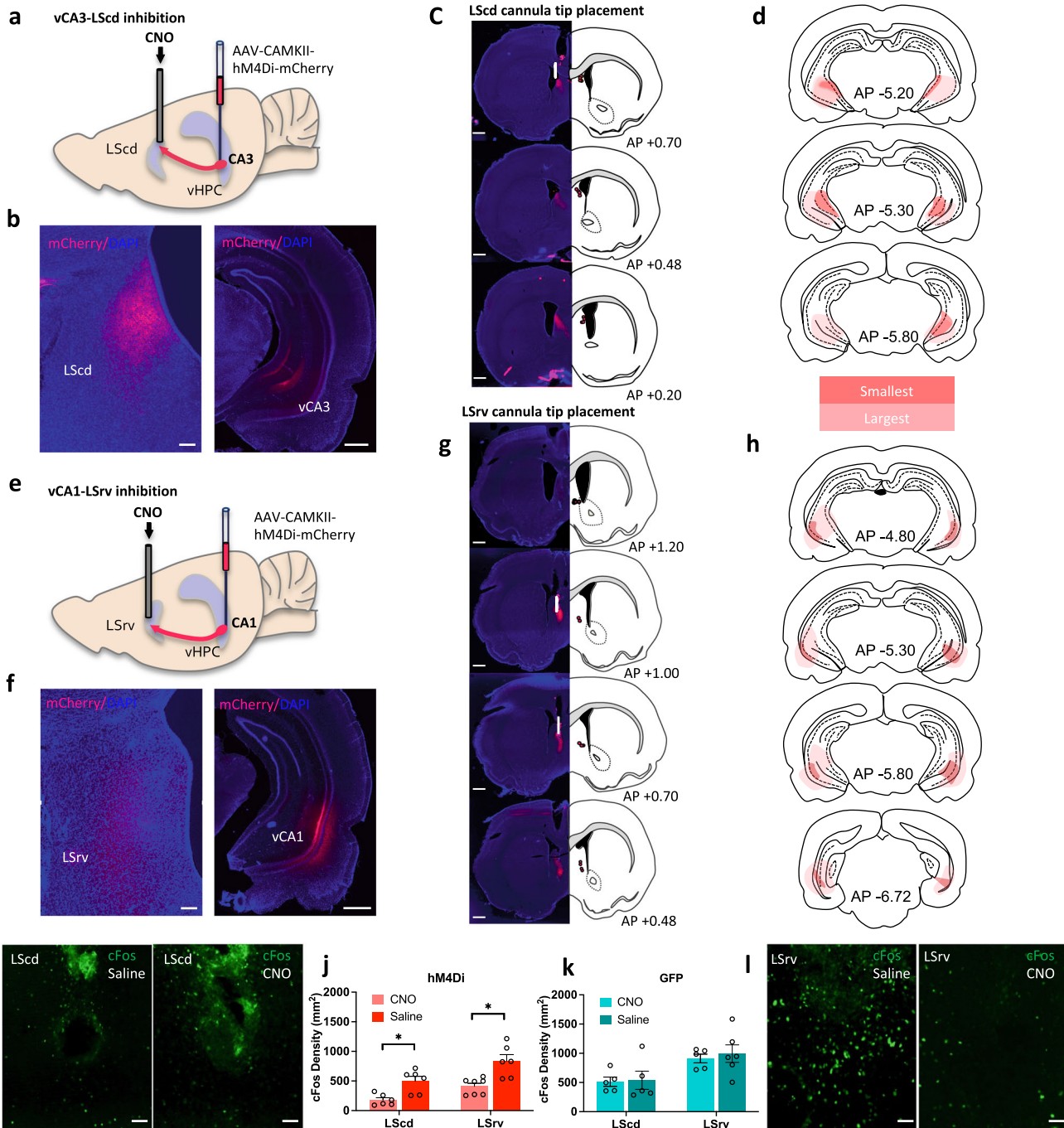

**Fig. 2 Experimental schematic and c-Fos results. a** Diagram of hM4Di transduction in vCA3 and cannulation in LScd, with **b** representative DREADDs axonal transduction in the LScd (red) and somatic transduction in vCA3, **c** cannula placements in the LScd restricted to sections +0.7 to +0.2 anterior to bregma[71], **d** minimum and maximum DREADDs expression in vCA3 observed in sections -5.20 to -5.80 posterior to bregma[71]. **e** Diagram of hM4Di transduction in vCA1 and cannulation in LSrv, with **f** representative DREADDs axonal transduction in LSrv and somatic transduction in vCA1, **g** cannula placement in the LSrv restricted to sections +1.2 to +0.48 anterior to bregma[71], and **h** minimum and maximum DREADDs expression in vCA1, observed in sections -4.80 to -6.72 posterior to bregma[71] **i**, **l** Representative images of c-Fos expression in the LScd and LSrv of hM4Di-expressing rats following either saline or CNO infusion. **j**, **k** Intracerebral CNO infusions attenuated c-Fos expression in areas of the LS that receive hM4Di positive inputs (Drug x Virus: $F_{(1,38)} = 5.17$, $p = 0.0248$; post hoc CA1 hM4Di $t_{7.16} = 3.49$, $p = 0.0022$; post hoc CA3 hM4Di $t_{7.21} = 3.64$, $p = 0.0228$; Region: $F_{(1,38)} = 25.73$, $p = 0.0001$) compared to saline controls. EGFP-LScd-CNO $n = 5$, EGFP -LScd-Saline $n = 5$, EGFP -LSrv-CNO $n = 5$, EGFP-LSrv-Saline $n = 6$. hM4Di-LScd-CNO $n = 6$, hM4Di-LScd-Saline $n = 6$, hM4Di-LSrv-CNO $n = 7$, hM4Di-LSrv-Saline $n = 6$. c-Fos analyzed by three-way ANOVA, followed by post hoc two-sided $t$-tests with Bonferroni-Holm correction for multiple comparisons. Data represent mean ± sem. *$p < 0.05$. * denotes between-subject comparisons. 100 um scale bars for **b**, **f** (left), **i**, **l** 1 mm scale bar for images **b**, **f** (right), **c**, **g**. **i** Cannula track shown by white vertical bar for images in **c** and **d**. dHPC dorsal hippocampus, LScd caudodorsal lateral septum, LSrv rostroventral lateral septum, vHPC ventral hippocampus. Source data are provided as a Source Data file.

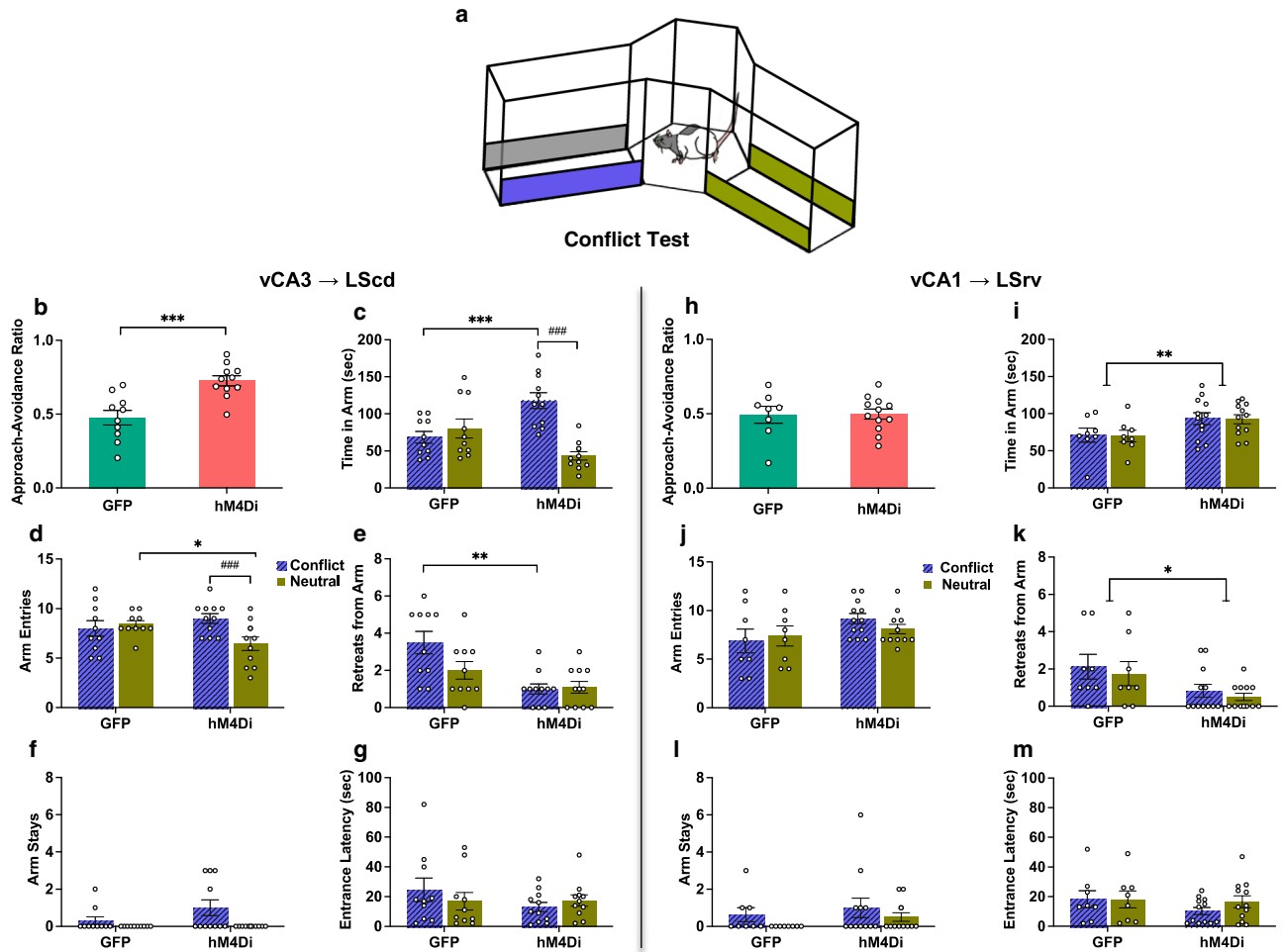

**Fig. 3 vCA3 → LScd and vCA1 → LSrv circuit inhibition led to divergent responses during conflict test. a** Schematic of conflict test configuration with grey and blue bars depicting the combined appetitive and aversive cues in one arm, and green bars representing the neural cues in another arm. **b**, **c** vCA3 → LScd hM4Di animals showed potentiated approach ($t_{16.18} = 4.17$, $p = 0.0005$) driven by spending more time in the conflict vs. neutral arm, and more conflict arm time than GFP controls (Arm x Virus: $F_{(1,19)} = 16.42$, $p = 0.0004$; post hoc arm $t_{10} = 5.74$, $p = 0.0001$; post hoc virus $t_{19} = 3.66$, $p = 0.0009$) **d** vCA3 → LScd hM4Di animals made more entries into the conflict than neutral, and less neutral arm entries than controls (Arm x Virus: $F_{(1,19)} = 8.89$, $p = 0.0160$; post hoc arm $t_{10} = 4.82$, $p = 0.0001$; post hoc virus $t_{19} = 2.44$, $p = 0.0392$), and **e** retreated less from both arms than the controls (Virus: $F_{(1,19)} = 15.76$, $p = 0.0009$; Arm x Virus: $F_{(1,19)} = 6.63$, $p = 0.0723$; post hoc virus conflict $t_{12.53} = 3.80$, $p = 0.0012$; post hoc virus neutral $t_{19} = 1.63$, $p = 0.0944$) **f** Both vCA3 → LScd groups stayed in the conflict more than neutral (Arm: $F_{(1,19)} = 6.98$, $p = 0.0161$), and **g** did not differ in latency to enter the arms (Lowest P: $F_{(1,19)} = 1.20$, $p = 0.2976$). **h** vCA1 → LSrv hM4Di animals had equal approach-avoidance ratios ($t_{11.96} = 0.06$, $p = 0.9548$), **i** spent more time in both arms compared to controls (Virus: $F_{(1,18)} = 10.54$, $p = 0.0047$), **j** had similar numbers of entries into the maze arms as the EGFP animals (Virus: $F_{(1,18)} = 2.46$, $p = 0.1340$), **k** and fewer retreats (Virus: $F_{(1,18)} = 6.10$, $p = 0.0220$). **l** Both vCA1 → LSrv groups trended toward more conflict arm stays (Arm: $F_{(1,18)} = 3.14$, $p = 0.0935$), **m** and no difference in latency to enter either arm (Lowest P: $F_{(1,18)} = 1.86$, $p = 0.1893$). Tests are unpaired two-sided t-tests (**b**, **h**) and two-way ANOVAs, followed by post hoc two-sided t-tests with Bonferroni-Holm correction for multiple comparisons (**c–g**, **h–k**). vCA3-hM4Di $n = 11$, vCA3-GFP $n = 10$, vCA1-hM4Di $n = 12$, vCA1-GFP $n = 8$. Data represent mean ± sem. *$p < 0.05$, **$p < 0.01$, ***$p < 0.001$. * denotes between-subject, and ♯ denotes within-subject comparisons. Source data are provided as a Source Data file.

explore the appetitive and neutral arms in one test, and the aversive and neutral arms in another (Fig. 4a, b). During the appetitive preference test all groups spent more time in the arm that signaled sucrose reward relative to the neutral arm (Fig. 4c, d) and chose to actively stay in the arm rather than leaving (Fig. 4e, f), confirming that neither of the vHPC → LS pathways are required for approach responses to the appetitive cue alone. Interestingly, while all groups showed relatively few retreats from either arm (Fig. 4g, h), the vCA1 → LSrv inactivated group showed significantly fewer retreats from the appetitive arm than their controls, suggesting that this pathway can exert some effects outside of conflict scenarios.

Conversely, during the cue avoidance test, all groups spent less time in the aversive arm compared to the neutral (Fig. 4i, j),

indicating that inactivation of neither the vCA3 → LScd nor vCA1 → LSrv pathway rendered animals less avoidant of shock-predictive cues. Animals, regardless of manipulation, also rarely chose to stay in the aversive arm following entry (Fig. 4k, l), supporting the idea that pathway manipulations did not make aversive cues more appealing. Finally, all groups consistently emitted a high number of retreats to the aversive arm (Fig. 4m, n) confirming that neither pathway manipulation altered cue-specific avoidance responses.

Collectively, the evidence suggests that neither vCA3 → LScd nor vCA1 → LSrv circuit inhibition alone has a significant effect on overall preference for stimuli with a single associated valence, and neither pathway affects memory or cue preferences under normal conditions.

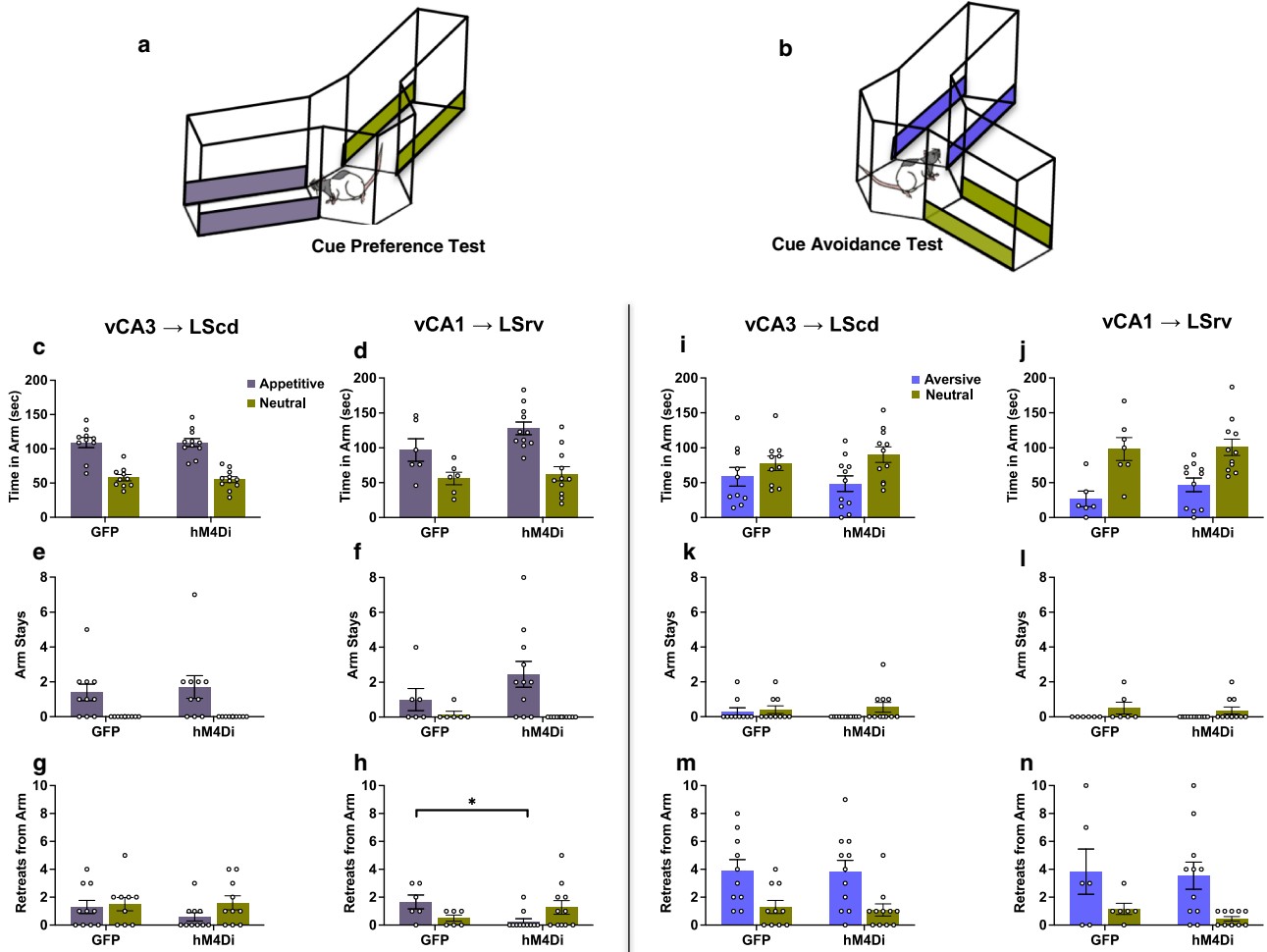

**Fig. 4 Chemogenetic inhibition of vHPC-LS pathways does not affect conditioned cue preference or avoidance tests. a**, **b** Diagrams show the configuration of a cue preference test in which animals were allowed to freely explore an arm imbued with appetitive cues (grey bars), and an arm with neutral cues (green bars), and cue avoidance test in which animals were presented with an arm with aversive cues (blue bars) and another arm with neutral cues (green bars). **c**, **d** Regardless of pathway targeted or virus type, animals spent more time in the appetitive arm than the neutral arm (vCA3 Arm: $F_{(1,19)} = 88.56$, $p = 0.0001$; vCA1 Arm: $F_{(1,15)} = 16.22$, $p = 0.0011$; vCA1 Arm x Virus: $F_{(1,15)} = 0.85$, $p = 0.3709$), **e**, **f** and emitted arm stays exclusively in the appetitive arm (vCA3 Arm: $F_{(1,19)} = 14.78$, $p = 0.0002$; vCA1 Arm: $F_{(1,15)} = 8.33$, $p = 0.0113$). **g** vCA3 → LScd groups did not differ in the number of retreats emitted to either arm (Arm: $F_{(1,18)} = 2.33$, $p = 0.1468$; Arm x Virus: $F_{(1,18)} = 1.04$, $p = 0.3205$), while **h** vCA1 → LSrv hM4Di animals retreated less from the appetitive arm than their controls (Arm x Virus: $F_{(1,15)} = 7.85$, $p = 0.0112$; post hoc virus appetitive $t_{6.60} = 2.62$, $p = 0.0338$). **i** The vCA3 → LScd groups spent less time in the aversive arm (Arm: $F_{(1,19)} = 4.88$, $p = 0.0395$). **j** vCA1 → LSrv groups both spent less time in the aversive arm (Arm: $F_{(1,15)} = 17.33$, $p = 0.0008$). **k**, **l** Stays were rarely emitted in either arm, and tended to occur in the neutral rather than the aversive (vCA3 Arm: $F_{(1,19)} = 2.11$, $p = 0.1624$; vCA1 Arm: $F_{(1,15)} = 5.40$, $p = 0.02548$), **m**, **n** All groups were more likely to retreat from the aversive arm than the neutral (vCA3 Arm: $F_{(1,19)} = 22.67$, $p = 0.0002$; vCA1 Arm: $F_{(1,15)} = 10.15$, $p = 0.0039$). vCA3-hM4Di $n = 11$, vCA3-GFP $n = 10$, vCA1-hM4Di $n = 11$, vCA1-GFP $n = 6$. Tests are two-way ANOVAs, followed by post hoc two-sided $t$-tests with Bonferroni-Holm correction for multiple comparisons. Data represent mean ± SEM. *$p < 0.05$. * denotes between-subject comparisons. Source data are provided as a Source Data file.

**Inhibition of the vCA3 → LScd decreases the latency to feed in a novel environment, while vCA1 → LSrv inactivation lengthens the time to bite.** Approach-avoidance conflict is a key feature of many anxiogenic situations that require searching for rewards and avoiding threats, such as foraging for food in a novel and potentially dangerous environment. This type of conflict is elicited during the novel environment suppressed feeding task (NESF), which introduces a conflict between the need to consume food and the need to monitor for threats in a novel environment. Before the NESF task, rats were food restricted overnight, and on the day of testing separated from their cage mates at least an hour prior to infusion, to allow them to habituate to single housing in a clean cage. Following CNO infusion, rats were transported to a novel testing room and placed in a square open field with a single food pellet near a side wall (Fig. 5a). During the NESF, rats

exhibited a reliable sequence of behaviors, first exploring their new environment prior to making an exploratory bite on the food pellet without initiating feeding. Animals then continued to explore the environment, or make very short contact with the food that lasted only a few (<2 s) seconds, before finally initiating continuous feeding. vCA3 → LScd circuit inhibition led to faster initiation of exploratory bites of the pellet (Fig. 5b), and subsequent initiation of continuous feeding (Fig. 5c), compared to the controls. However, it was noted that the difference in latency between the exploratory bite and feeding initiation was similar between the vCA3 → LScd circuit inhibition and control groups (Fig. 5d), suggesting that this circuit exerts its effects primarily during the initial food approach during environmental exploration. If the experimental manipulation had exerted its effect by increasing the hunger drive, one would expect that the difference

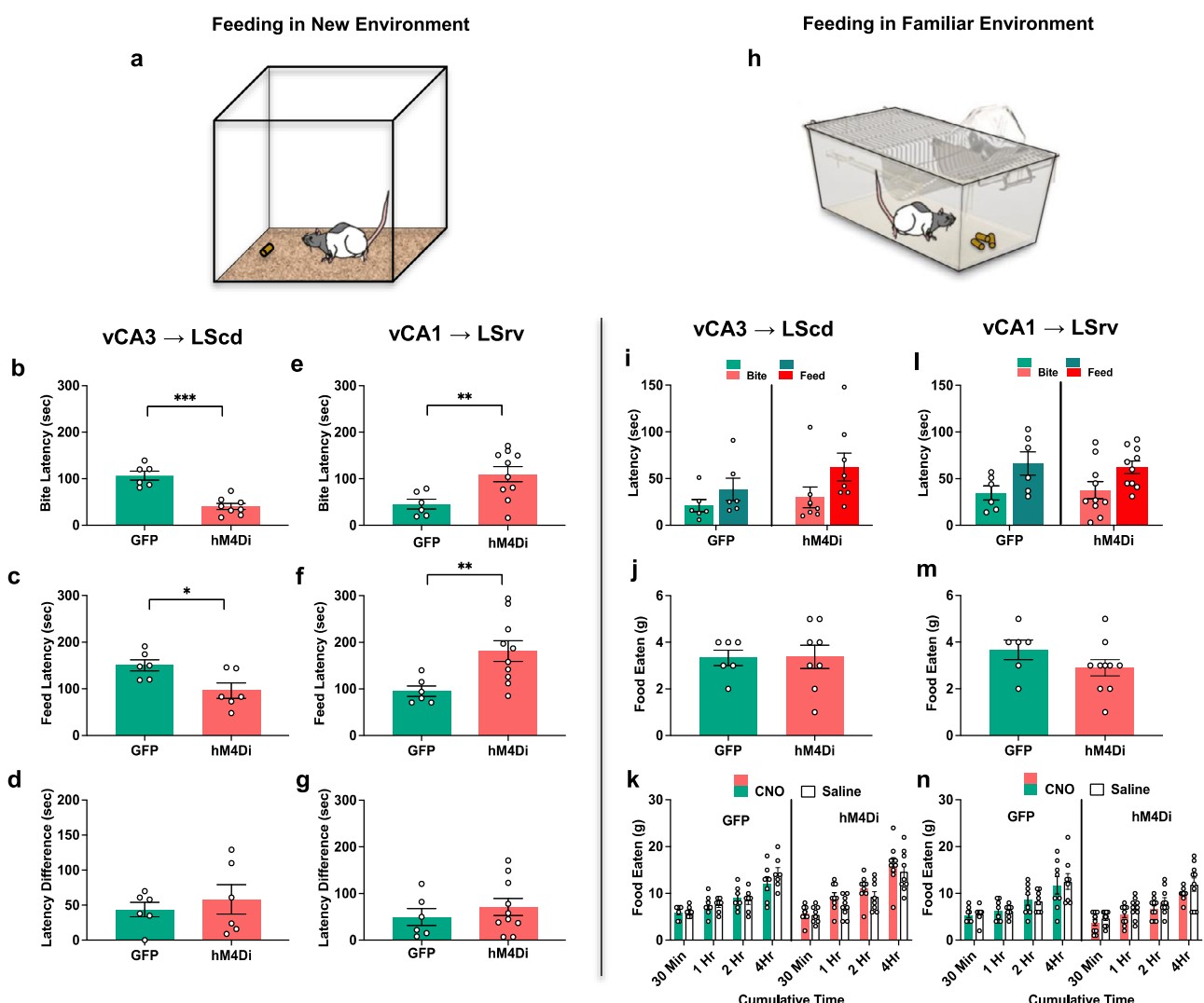

**Fig. 5 vHPC → LS circuits exert differential effects on food approach in a novel environment, but not familiar environment. a** NESF test schematic. **b** vCA3 → LScd hM4Di animals were faster to bite the food pellet in the novel environment ($t_{12} = 5.94$, $p = 0.0001$) and **c** faster to start feeding ($t_{10} = 2.70$, $p = 0.0214$) compared to GFP controls, **d** while both groups had similar latencies from bite to feeding ($t_{10} = 0.62$, $p = 0.5349$). **e** vCA1 → LSrv animals were slower to bite the pellet in the novel environment ($t_{13.7} = 3.45$, $p = 0.0097$) and **f** slower to start feeding ($t_{12.7} = 3.31$, $p = 0.0050$) compared to GFP controls, while **g** both groups had similar latencies from bite to feeding ($t_{14} = 0.78$, $p = 0.4379$). **h** Familiar environment feeding schematic. **i, l** In familiar environments all groups had similar latencies to bite (vCA3 virus: $t_{12} = 0.62$, $p = 0.6845$; vCA1 virus: $t_{14} = 0.23$, $p = 0.8168$) and start feeding (vCA3 virus: $t_{12} = 1.19$, $p = 0.2622$; vCA1 virus: $t_{14} = 0.31$, $p = 0.7575$), and **j, m** consumed similar amounts of food in the familiar environment feeding test (vCA3 virus: $t_{12} = 0.06$, $p = 0.8233$; vCA1 virus: $t_{14} = 1.38$, $p = 0.1691$). **k** No difference between vCA3 → LScd groups, with both consuming more food as time progressed in home cage feeding test (Time Bin: $F_{(3,42)} = 90.29$, $p = 0.0001$). **n** Both vCA1 → LSrv groups consumed more food over time in the home cage feeding test (Time Bin: $F_{(3,48)} = 72.05$, $p = 0.0001$) and consumed more food overall after CNO, as compared to saline (Drug: $F_{(1,16)} = 5.11$, $p = 0.0427$; Drug x Bin: $F_{(3,48)} = 0.6579$, $p = 0.6494$). NESF and FESF: vCA3-hM4Di $n = 8$, vCA3-GFP $n = 6$, vCA1-hM4Di $n = 10$, vCA1-GFP $n = 6$, unpaired two-sided $t$-tests with Bonferroni-Holm correction for multiple comparisons. Home cage feeding: vCA3-hM4Di $n = 9$, vCA3-GFP $n = 7$, vCA1-hM4Di $n = 10$, vCA1-GFP $n = 8$, three-way ANOVAs with Bonferroni-Holm correction for multiple comparisons. Data represent mean ± SEM. *$p < 0.05$, **$p < 0.01$. * denotes between-subject comparisons. Source data are provided as a Source Data file.

in latency between the exploratory bite and continuous feeding would be shorter in the vCA3 → LScd inhibition group, since the need to satiate hunger may dominate over the need for environmental exploration, but this was not observed in our study. In contrast, inactivation of the vCA1 → LSrv circuit led to a longer latency to make an exploratory bite on the food pellet (Fig. 5e) as well as initiating feeding compared to CNO infused control animals (Fig. 5f). Similar to the above findings, the time difference between the exploratory bite and initiating feeding was similar between groups (Fig. 5g), suggesting that the manipulation primarily affected behavior upon initial exposure to the food/

environment, suggesting that this circuit plays a role in early environmental exploration.

**Effects of circuit inactivation cannot be accounted for by changes to basic feeding behavior, novelty detection, non-specific anxiety, or locomotion.** A growing body of literature has implicated the vHPC → LS circuits in the regulation of feeding behaviors, even in the absence of environmental conflict[28,42]. Since the forms of motivational conflict that we found to be susceptible to vHPC → LS inhibition involved food in some

capacity, it is possible that alterations in basic feeding processes could explain these findings. To test for this possibility, immediately following the NESF, rats were placed within their familiar cages with 20 g of pellets and allowed to consume them freely for 10 min (Fig. 5h). Animals were not given the opportunity to ingest the food at great length (<5 s) during NESF to ensure they would be sufficiently hungry prior to the feeding test in a familiar environment. In contrast to the effects seen in the NESF, in a more familiar environment, neither vCA3 → LScd (Fig. 5i) nor vCA1 → LSrv (Fig. 5l) inactivation affected the latency to bite or feed relative to controls. Rats also ate a similar amount of food in all groups (Fig. 5j, m), suggesting that these circuits are important for the initial approach toward food in a novel environment rather than food consumption in general. To further examine the effect of circuit inhibition on feeding behavior on a longer timescale, both vCA3 → LScd and vCA1 → LSrv animals were allowed to freely feed in their home cages following either CNO or saline infusions, and the cumulative amount of food consumed 30 min, 1, 2, and 4 h afterward was measured. The same test was repeated the following day after animals received the opposite infusion type. The vCA3 → LScd animals (Fig. 5k) consumed similar amounts overall and fed at approximately equal rates, with no significant differences in food consumption detected at any time point. For the vCA1 → LSrv groups (Fig. 5n) a significant effect of drug was detected across both virus groups, although the effect of drug on food consumption was not significant at any individual time bin, and overall, the cumulative pattern of feeding was unchanged. Analyzing the data by the amount of food consumed within an individual time bin rather than cumulatively also failed to show any significant effect of pathway inhibition for either vCA3 → LScd or vCA1 → LSrv animals (Supplementary Fig. 2). Thus, the effect of CNO mediated inactivation of both the vCA3 → LScd and vCA1 → LSrv circuits on food consumption fails to adequately explain the effects observed in either the Y-maze conflict test or the NESF.

Additionally, to ensure that inactivation of the vHPC → LS circuits did not lead to changes in novelty detection or preference given the relatively novel configuration of cues in the conflict test and the novel environment in the NESF, we habituated animals to two arms of the Y-maze with access to distal spatial cues, as well as non-spatial wall patterns, followed by a second maze exposure with a third arm open in a novel spatial location and with a visually distinct wall pattern. Regardless of circuit targeted or virus type, all groups of rats showed a similar increase in time spent in the novel arm (Supplementary Fig. 3a, b), signaling successful detection of the novel features of the third arm relative to the average preference for the two familiar arms, and suggesting that vHPC → LS circuit inactivation does not affect novelty detection or preference per se.

In addition to the Y-maze conflict and NESF test, approach-avoidance conflict is also induced by exploration of a new area with aversive features, which can be tested in rodents with the elevated plus maze (EPM), a common test for anxiety in rodents. To examine whether anxiety-like behavior was impacted by circuit inhibition, animals were placed in the EPM and allowed to explore the apparatus for 10 min. Rats in the vCA3 → LScd (Supplementary Fig. 3c) and vCA1 → LSrv (Supplementary Fig. 3d) groups demonstrated a preference for the closed arm, and neither vCA3 → LScd nor vCA1 → LSrv circuit inhibition led to significant changes in EPM performance, suggesting that neither pathway is important for the regulation of this type of approach-avoidance conflict.

Finally, since the behavioral disinhibition seen with circuit manipulation could be explained by an increase in non-specific locomotor activity, we examined animals' locomotor activity for 1 h in a clean empty cage without any motivational stimuli

present. Rats in all groups demonstrated typical locomotor patterns, with the distance traveled highest in the first 5 min and declining over a 1 h period, with no difference in any time bin nor overall difference in activity patterns by the circuit inactivated groups (Supplementary Fig. 3e, f). This and the above experiments indicate that the effects of circuit manipulation were only observed in the presence of motivationally conflicting stimuli.

*Differential downstream projections from the LS.* To further evaluate which downstream targets of the LS could potentially mediate the behavioral effects induced by vHPC → LS circuit inhibition, a separate set of animals received AAV1-CAG-GFP infusions into either the LScd (Supplementary Fig. 4a, b) or LSrv (Supplementary Fig. 4i, j). For LScd animals, reliable projections were noted in the horizontal diagonal band, peduncular lateral hypothalamus, supramammillary nucleus, and ventral tegmental area (VTA) (Supplementary Fig. 4c–h). For LSrv animals, projections were observed in the septohypothalamic area, the medial preoptic area, anterior hypothalamic areas, ventromedial hypothalamic areas, dorsomedial hypothalamic area, and tuberal lateral hypothalamus (Supplementary Fig. 4k–p). The observed results fit with the general pattern observed in other studies[17,27], with the LScd projecting to more lateral parts of the hypothalamus and posterior regions such as the supramammillary nucleus and VTA, while the LSrv targets more medial components of the hypothalamus.

## Discussion

The vHPC is a critical mediator of approach-avoidance conflict resolution[1], and the LS has been implicated in the regulation of behavior in both positively and negatively valanced situations[43–45]. Our work bridges these two streams of findings to demonstrate that parallel, regionally specific circuits connecting the vCA3 and vCA1 to the LS play critical, and dissociative roles in approach-avoidance conflict resolution. During learned approach-avoidance conflict resolution, avoidance responses elicited by conflict stimuli are primarily under the control of the vCA3 → LScd circuit, mediating avoidance of stimuli that predict both positive and negative outcomes. Inhibition of this circuit results in a behavioral profile similar to those seen with lesions of the entire vHPC[11,20], as well as direct inactivation of the vCA3 and closely associated vDG[12,13]. Interestingly, the vCA1 → LSrv appears to control distinct aspects of behavioral inhibition, as this circuit decreases approach responses to both conflict and neutral stimuli during learned approach-avoidance conflict. These results only partially replicate the result of direct vCA1 inactivation seen in the Y-Maze conflict task, which led to an increase in the amount of time spent in the neutral arm but not the conflict arm[12]. This suggests that the vCA1 regulates approach responses during conflict through multiple pathways, with the vCA1 → LSrv circuit important for suppressing non-specific approach responses to contextually inappropriate stimuli. While the distinct parallel pathways from the vCA3 and vCA1 to the LS have long been recognized[17], this study demonstrates a functional dissociation between the two circuits.

One point of discussion is whether the observed potentiation in conflict cue-elicited approach behavior in the vCA3-LScd inhibited group was a manifestation of an increase in approach bias or a decrease in avoidance bias. These possibilities cannot be readily disentangled based on the present behavioral data, and it may be more productive to interpret the current data in the light of the suggested role of the vHPC in behavioral inhibition[5]. According to this viewpoint, the vHPC responds to conflicting goals by sending a global excitatory signal to the rest of the brain to increase the salience of negative stimuli or suppresses appetitive

associations, contributing to a general state of anxiety[1,5,46]. Our results are in general agreement with this account, but provide novel mechanistic insight into the proposed global mechanism of affective modulation of behavior undertaken by the vHPC. More specifically, our data suggest that the vHPC serves as a key hub in a system responsible for inhibiting behavioral control systems in the presence of motivational conflict, via discrete downstream targets in the LS (i.e., caudodorsal vs. rostroventral). The vast majority of LS projection cells are GABAergic and inhibit their own downstream targets in turn[26,47], and the midline and lateral hypothalamic nuclei are some of the most prominent targets of the LSrv and LScd respectively[17,48,49]. The medial hypothalamus, along with its connection to the dorsal periaqueductal gray mediates escape/avoidance[50] alongside a variety of social behaviors[44,51], while the lateral hypothalamus mediates innate behaviors that are important for homeostatic regulation such as consummatory responses[52], as well as more general behavioral excitation[53]. The LS is therefore in a prime position to regulate a diverse set of hypothalamus-mediated behaviours via inhibitory transmission, and may be recruited by the vHPC when situations activate distinct behavioral programs that cannot both be emitted simultaneously, such as reward approach and threat avoidance.

Our data suggest a nuanced contribution of the two HPC-LS pathways in the regulation of motivational control systems. The behavioral effects we observed in the vCA3-LScd inhibition group in the Y-maze conflict test (increased time spent, preference for entry into the conflict arm, and decreased retreat behavior) may be interpreted collectively to reflect the inability of these animals to inhibit their approach toward reward-related stimuli specifically in the face of a threat of an aversive event. Similarly, the observed decrease in the latency to make an exploratory bite on the pellet, rather than in the latency to commence feeding in the NESF test in the vCA3 → LScd circuit inhibition group may reflect disinhibited approach responses toward foodstuff in a novel, potentially threatening environment. Together, our findings suggest that the vCA3 → LScd circuit is recruited to inhibit food-related approach responses in the presence of motivational conflict, consistent with previous reports of the LS and HPC (albeit separately) in regulating the latency to feed in the presence of novel, potentially dangerous environments[20,24,40,54].

In contrast, we propose that the vCA1 → LSrv circuit is important for suppressing non-specific/general approach and exploration responses in favor of approach toward the most motivationally salient stimuli. Accordingly, inhibition of this circuit led to longer times in engaging with the pellet in the NESF test (as a product of increased exploratory behavior of the apparatus) and greater non-specific engagement with both the neutral and conflict arms in the Y-maze conflict test. Thus, in contrast to the traditional view that longer bite and feeding times in NESF tests are indications solely of greater anxiety[55,56], we suggest that they reflect disinhibited exploration of the environment at the cost of goal exploration. This is also consistent with work demonstrating that LS projecting vCA1 cells mediate focused search at the spatial location where food was previously found (under extinction conditions), in the absence of any memory impairment of the food location when food is present[57]. Altogether, the results suggest that the primary contribution of the vHPC on tasks like the NESF would be in the anticipatory responses to food rather than consumption, and the relationship between food approach and HPC to LS circuits may also explain alterations in other food-related behaviors seen following whole HPC lesions, which include animals approaching food receptacles more without consuming more food than controls[58,59], heightened conditioned locomotor activity during food anticipation[60], and a cessation of food hoarding[61].

We found little evidence that the vHPC to LS circuit is involved in feeding behavior in the absence of motivational conflict, despite a growing literature suggesting that the circuit is directly involved in food consumption even in neutral contexts[28,29,42,57]. While our study specifically targeted the vHPC to LS terminals, many of the studies that have found effects on food consumption have manipulated vHPC cells that project to the LS, a technique that also affects collateral projections of these cells to non-LS areas. A recent study found that LS projecting vCA1 cells are the most likely to have collateral projections to other regions, particularly those important for feeding such as the lateral hypothalamus and nucleus accumbens[25]. Thus, long term alterations in food consumption, seen during manipulations that affect LS projecting vCA1 soma, might be mediated through non-LS pathways. In fact, when studies have used loss of function manipulations of vHPC to LS circuits or directly manipulated the LS, the effects on food consumption tend to occur within the first 15 to 30 min but are less noticeable as time passes[28,62,63], consistent with the suggestion that the circuits are more influential during the initial approach toward food rather than on consummatory responses. Considering that the strongest effects that were elicited by pathway manipulations appeared during tasks that utilized food as a motivator, it is worth considering the extent to which these circuits are involved in conflict behaviors that do not involve food stimuli specifically. In principle, the HPC and LS have been found to mediate and project to regions involved in more complex reward processing, such as conditioned drug responses and socialization[43,64], and human conflict studies that have used money or game points as a motivator have observed HPC activation[14,15], suggesting that these systems should subserve a variety of motivational conflicts.

Our work consolidates the role of the vHPC subregions in the control of distinct aspects of approach-avoidance conflict resolution, which is a critical process that is engaged during environmental threats in both humans and animals and is thought to mediate decision making during anxiety[1,3,5,15]. Additionally, while parallel pathways connecting different HPC subfields to the LS have been long known[17,19], this study establishes a causal and functional link between discrete regions of the vHPC and LS in regulating approach-avoidance behaviors in the face of motivational conflict. We have identified the vCA3 connections to the LScd, which in turn send second order connections to downstream targets such as the lateral hypothalamus that may be critical in promoting avoidance in the face of motivational conflict, and the vCA1 to LSrv pathway to be involved in facilitating more general behavioral inhibition, which in turn projects to a wider variety of midline hypothalamic structures. Furthermore, given that the LS is implicated in a rich repertoire of behaviors beyond those investigated here, including stress induced grooming and social behavior[38,51,65], future work should elucidate the ways the identified circuits might work to arbitrate between behaviors outside of traditional and explicit approach-avoidance conflict paradigms, and their relationship with psychiatric disorders associated with the dysregulation of behavioral inhibition[1–3].

## Methods

**Subjects**. Sixty-three male Long-Evans rats weighing 300–400 g at the time of surgery were used (Charles River Laboratories, NJ, USA). They were housed in groups of two in a room held at a constant temperature of 21 degrees, under a 12 h light/dark cycle (lights on at 0700 h). Water was available ad libitum, while food was restricted to ~19 g of lab chow per day prior to behavioral testing, sufficient to maintain 85–90% of pre-testing body weight. All experiments were conducted during the light phase for 6 days/week, and in accordance with the guidelines of Canadian Council of Animal Care and approved by the University and Local Animal Care Committee of the University of Toronto.

**Stereotaxic surgery**. Animals were anesthetized with isoflurane (Benson Medical, ON, Canada) and placed in a stereotaxic frame (Steolting Co, IL) with the incisor bar set below the interaural line. An incision was made along midline of the skull and the skin retracted to reveal the bregma. Burr holes were drilled at the transduction and cannulation sites. Fifty-seven animals received bilateral infusions of either AAV8-CAMKII-hM4Di-mCherry or AAV8-CAMKII-EGFP control viruses (Addgene, MA) into either the ventral CA3 (AP: −5.10 mm, ML: ±5.10 mm, DV: −7.20 mm) or CA1 (AP: −5.80 mm, ML: ±5.60 mm, DV: −7.30 mm), with 0.5 µl for each infusion. Bilateral guide cannulae (26 gauge; Plastics One, Roanoke, VA) were bilaterally implanted into either the LScd (AP: +0.20 mm, ML: ±0.80 mm, DV: −3.00 mm), or LSrv (AP: +0.50 mm, ML: ±0.80 mm, DV: −4.50 mm) with the tips of the cannulae sitting 1.5 mm above the target coordinates. LS coordinates were chosen to avoid simultaneous vCA3 and vCA1 terminal inhibition in the same animal by ensuring an adequate distance between the sites where the subfield projects to. The cannulae were fixed to the skull with dental cement and four jewellers' screws. Solid stainless-steel dummy cannulae (Plastics One) were inserted into the guide cannulae following surgery and were covered with dust caps to prevent their accidental removal. Rats were given at least 7 days to recover before the start of behavioral training, and training was timed to ensure that the hM4Di receptors would have at least 6 weeks to express before attempted activation.

An additional six animals were used for the anterograde tracing study, with four animals receiving unilateral infusions of AAV1-CAG-GFP into the LScd (AP: +0.20 mm, ML: ±0.75 mm, DV: −4.70 mm) and two animals receiving infusions of the in the LSrv (AP: +0.50 mm, ML: ±0.80 mm, DV: −5.50 mm). These animals were sacrificed at least 8 weeks following surgery.

**Intracranial microinfusions**. In order to minimize the mechanical effects of subsequent drug infusions and habituate the animals to the infusion procedure, animals underwent an initial saline infusion at least 24 h before the conflict test. On critical test days, animals received 0.3 µl of CNO (1 mM, CNO-dihydrochloride, BioTechne, MN) or 0.9% saline vehicle (only for homecage feeding tests and EPM/c-Fos) into the LS target site. The substance was infused at a rate of 0.3 µl/min for 1 min via 30-gauge microinjectors (33 gauge; Plastics One) projecting 1.5 mm below the indwelling guide cannulae using an infusion pump (Harvard Apparatus, Hollison, MA) mounted with 10 µl Hamilton syringes. The microinjectors were left in place for a further 1 min to allow the drug to diffuse away from the injector tip. During microinfusion, animals were held and carefully handled to minimize stress and avoid equipment damage due to movement. The relevant test was administered a minimum of 10 min after the end of each infusion. Rats received six separate microinfusions following infusion habituation, which occurred on separate days prior to the conflict test (followed by locomotor test), cue preference/avoidance tests, novelty detection (followed by NESF), both free feeding tests, and EPM (Fig. 1).

**Behavioral procedures**. Following recovery from surgery, animals were administered a series of behavioral tests as shown in Fig. 1. Where appropriate, we conducted multiple tests on the same day of drug/saline microinfusion, to minimize the total number of repeated intra-cerebral infusions given to the animals ($n = 6$). Animals began their training in the mixed valence Y-maze task first. Following a successful cue acquisition test, animals were administered a conflict test, prior to which animals received microinfusions of CNO (1 mM). Approximately 1 h following the conflict test, the animals were transferred to another testing room to undergo the locomotor test. The next day, animals received a refresher conditioning session, and the day following that were administered the appetitive and aversive preference tests in counterbalanced order, with the first test beginning 10 min following drug infusion.

Following the completion of the Y-maze experiment, animals were given free access to food in their home cages for at least 5 days, to ensure they were not under chronic food restriction conditions prior to subsequent feeding tests. Food was then removed again over 18 h, and animals underwent another drug/saline microinfusion, followed by a novelty detection test, NESF and the familiar environment suppressed feeding test (FESF) immediately after.

Following these tests the animals were given free access to food for 24 h, and were then food restricted again prior to the two free feeding tests, which occurred on successive days. Animals received another two sets of infusions on those days (CNO and saline). Finally, animals received their final infusion of CNO or saline, and were administered the EPM, and sacrificed thereafter.

Variations in the reported number of animals used in different phases of the study (Fig. 1) were due to unexpected data exclusions and criteria established a priori. The former included illness at various points in the study ($n = 4$) and technical/experimenter error during data collection ($n = 2$ for preference/avoidance tests, $n = 3$ for NESF/FESF tests, $n = 6$ for immunohistochemistry), and the latter included animals failing to reach criterion learning in the Y-maze task ($n = 12$), misplacement of cannulae or viral expression ($n = 4$), or animals failing to initiate contact with the food in the 10 min NESF test ($n = 1$).

## Mixed-valence Y-maze task

*Maze apparatus*. Behavioral testing for the approach–avoidance conflict task took place in a six-arm radial maze (Med Associates, VT) placed on a rotatable table

elevated 80 cm from the floor. The maze consisted of six enclosed arms (45.7 cm [L] × 16.5 cm [H] × 9.0 cm [W]) stemming from a central hub compartment with six automatic stainless-steel guillotine doors allowing access to the arms. Arms were enclosed by Plexiglas walls and a removable Plexiglas lid, and contained a stainless grid floor connected to a shock generator (Med Associates). The entire maze was covered in red cellophane paper to block visibility of any extramaze cues, while enabling video recording of behavior via a video camera mounted above the apparatus. The end of each arm contained a receding well consisting of a stainless-steel tray that could be connected to a syringe pump for the delivery of liquid sucrose. Only three out of six arms were used at any one time in an experimental session, forming a Y-maze. The maze was wiped down with 70% ethanol solution after each session to eliminate odor traces, and the maze was randomly rotated left or right between days by varying degrees (60, 120, or 180) to minimize possible conditioning to intramaze cues or relative spatial location of the arms.

*Habituation*. Training began with four sessions to habituate animals to the maze apparatus. For session one, animals were placed in the central hub for 1 min, after which all three doors were lifted allowing the animal to explore 3 of the maze's arms for 5 min. The arms contained no cues. After 5 min the doors closed, and the animals were returned to the hub through a manually opened door before being removed from the maze. In the second habituation session animals were exposed to 3 sets of visuotactile cues (vinyl, duct tape, or wooden textured bars), with each set consisting of two rectangular bars (45 cm [L] × 4 cm [W] × 0.5 cm [D]) attached to the sides of the arm walls. Animals were allowed to explore the Y-maze arms containing these cues for 5 min following a 1 min period in the hub. Following this session cues were designated appetitive, aversive, or neutral, as determined by the amount of exploration time in the habitation session: the most preferred cue set was designated the aversive cue, the least explored the appetitive cue, and the remaining one the neutral cue. The third habituation session consisted of a two-arm maze configuration, with one arm containing a combination of one of the "appetitive" cues and one of the "aversive" cues placed on opposite walls, and a separate arm containing neutral cues. This session mimicked the conditions of the conflict test, and controlled for the effects of novel combinatory stimuli during the conflict test. In the fourth session animals were re-exposed to the maze and confined to one arm at a time for 2 min to habituate them to the cue conditioning procedure but without the presence of any cues.

*Cue-outcome conditioning*. Animals underwent daily conditioning sessions to acquire the appetitive, aversive, and neutral cue-outcome associations. Each session began with 30-s adaptation in the hub, followed by 2 min of confinement in each of the three arms. The order of arm presentation was varied daily to prevent the animals associating the outcomes with the sequence of arm presentation. In the arm containing the appetitive cues the animals were given four aliquots of 0.4 ml 20% sucrose solution delivered in 20 s intervals. In the arm with the aversive cues the animals received four mild shocks (~1 s duration, mean = 0.36 mA; range = 0.30–0.42 mA) delivered at random intervals, beginning 15 s after entry and on average every 30 s afterwards. Shock levels were individually calibrated per animal to elicit startle and back treading responses, but not freezing. In the neutral arm the animals did not experience any sucrose or shock outcomes. Previous studies from our laboratory have demonstrated that these reinforcement schedules and magnitudes of sucrose/shock facilitate the development of appropriate conditioned approach and avoidance responses without leading to generalized fear of the apparatus or freezing to the aversive cue, and results in balanced approach-avoidance ratios on average for control animals during the conflict test[11–13]. The specific bar cues the animals were initially exposed to were used throughout all training and subsequent testing.

After every fourth conditioning session, rats underwent a conditioned cue acquisition test to assess learning of the cue contingencies. In this test, rats freely explored the arms containing the appetitive, aversive, and neutral cues for 5 min in extinction, and the time spent in each of the arms was recorded. All sessions were recorded for offline analysis, during which additional behaviors were measured (See statistical analysis section). Successful acquisition of conditioned cue preference and avoidance was indicated by: (1) more time spent in the presence of the appetitive cue than the neutral or aversive cues (conditioned cue preference), in addition to (2) less time in the presence of the aversive cue than the appetitive and neutral cues (conditioned cue avoidance). Animals were given at least two acquisition tests before they were allowed to proceed to the conflict test. Those that did not learn by the second acquisition test were given an additional four training sessions and a third acquisition test. Upon acquisition of the cue-outcome associations, animals that succeeded in reaching criterion performance were given an additional day of retraining before being moved onto the conflict test.

*Mixed valence cue conflict test*. Procedures for the conflict test were identical to the third habituation session described above. During this 5 min session under extinction conditions, a conflict situation created by placing two stimuli signaling opposing outcomes (reward- or shock-associated cue) in one arm and presenting the neutral cues in another arm. All sessions were recorded for offline analysis.

*Appetitive and aversive preference/avoidance tests*. Following one session of retraining on the cue-outcome contingencies, animals received another

microinfusion and underwent a conditioned cue preference and conditioned cue avoidance test to assess conditioned approach and avoidance behavior in the absence of conflict. Animals received both tests in counterbalanced order, with tests occurring ~45 min apart. In the conditioned cue preference test, animals allowed to freely explore (for 5 min) two arms, one containing the neutral cue, and the other containing the appetitive cue. In the conditioned cue avoidance test, animals were presented with the aversive cues in one arm, and neutral cues in the other. All sessions were recorded for offline analysis.

**Novel environment suppressed feeding (NESF) and familiar environment feeding**. Food was taken away from animals at least 18 h prior to testing, to ensure animals were motivated to eat during the task. Animals were separated into individual clean cages with access to water at least 2 h prior to infusions to allow for habituation to the "familiar" environment. Following infusions and a waiting period, animals were transported to a novel procedure room for the commencement of the task. A single Noyes pellet was placed along the far wall of a novel environment consisting of a square open arena made of clear plexiglass (45 cm [L] × 45 cm [W] × 40 cm [H]), with white paper lining the base of the arena on the outside and black coverings on the walls. An arena of this size was chosen to ensure that the animals found the pellet quickly and any differences in consumption time were not due to animals being unaware of its presence. The animal was placed in the arena on the wall opposite the pellet, facing the center. The animal was allowed to explore the environment freely until it began to consume the food pellet for a single feeding bout lasting 5 s, after which it was promptly removed from the maze. The task was terminated after 10 min if this condition was not met. Contact time with the food and the initiation of feeding were recorded throughout for offline analysis.

For familiar environment feeding, 20 g of food in a petri dish was placed at one end of the familiar cage. Immediately upon completing the NESF task animals were placed on the opposite end of the cage from the food. Animals were left alone in the room and allowed to freely feed for 10 min. At the end of the task, all food was removed and weighed to determine how much was consumed. Behavior was recorded in the familiar cage via camera through the side of the cage.

**Home cage free feeding task**. Animals were food restricted for at least 18 h before the free feeding task. One hour prior to testing (~1000–1100 h) animals were separated from their cage mates into individual clean cages. Cage mates were taken in their separate cages to the infusion room, where one animal received either CNO or saline, and the other received the opposite. After infusion, animals were returned to their standard vivarium housing room and given 12 standard Noyes pellets (~45 g) in their feeding hoppers. Food consumption was measured 30 min, 1, 2, and 4 h post feeding. Following the end of the first test, paired animals were rehoused together overnight on food deprivation. Animals that consumed <15 g during the free feeding test were given extra pellets to stabilize their weight loss for the remainder of the test. The following day the procedure was repeated with each animal's drug assignment counterbalanced, after which animals were rehoused together on free feed.

**Novelty detection test**. To assess whether circuit inhibition affected animals' responses to novel stimuli, a novelty detection test was administered in a Y-maze with access to both spatial and non-spatial cues. The walls of each maze arm were decorated with one of three distinct visual cues (black dots, diagonal stripes, and horizontal stripes), and extramaze cues were visible through transparent arm lids. During an initial habituation period two maze arms were opened, and rats were placed at the end of one arm facing toward the center of the maze. The rats explored the apparatus without disruption for 10 min, after which they were removed and placed back in their home-cage for 30 min prior to the novelty test. During the test phase, rats were given access to a third "novel" arm and the two familiar arms for 5 min. The time spent exploring each arm was recorded, and an average of the time spent exploring the two familiar arms was calculated for comparison with the time spent in the novel arm.

**Locomotor activity**. To control for alterations in general activity levels a locomotor test was administered. Animals were placed into individual opaque cages [44 cm (L) × 24 cm (W) × 20 cm (H)] lined with standard bedding material and topped with stainless steel cage lids in a quiet room. Animals were left for 1 h while their locomotor behavior was monitored using an overhead camera and processed by EthoVision tracking software (Noldus Information Technology, ON, Canada). Total distance traveled (in cm) was divided into 5 min time bins, for a total of 12 individual bins.

**Elevated plus maze**. After completion of all other tests, animals were infused with either CNO or saline as described above (n = 23 animals received saline as c-Fos controls—the behavioural data for these animals are not shown), and subjected to the EPM, a commonly used test of innate anxiety, which has been shown to recruit and depend upon both the vHPC[21,24] and LS[34,56]. The maze was composed of gray Perspex, with a central platform [10 cm (L) × 10 cm (W)] that connects four arms, two of which are enclosed in walls [40 cm (L) × 10 cm (W) × 22 cm (H)]. Rats were placed in the central platform facing an open arm and allowed to explore the maze for 10 min. The time spent in arms and the center and arm entries were measured.

**Histology**. Animals were injected with a lethal dose of Euthanyl (2 ml/4.5 kg; Bimeda, Cambridge, ON) and perfused intracardially at the ascending aorta with phosphate-buffered saline, then a 4% paraformaldehyde solution. For animals whose brains were later analyzed for c-Fos, this process occurred 70 min following the end of the EPM, to ensure that c-Fos activity at the point of paraformaldehyde fixation occurred 90 min after the midpoint of the EPM. Brains were harvested and kept in 4% paraformaldehyde for 24 h, after which it was transferred to 0.01% sodium azide.

**c-Fos immunohistochemistry**. In total, 50 μm coronal sections were sliced and collected using a vibratome (Leica, VT1200S) and first treated with 1% hydrogen peroxide for 30 min, then incubated in 0.5% blocking buffer (Roche, Mannheim, Germany—Cat 11096176001) in 1x Tris/NaCL buffer for 1 h at room temperature. Sections were incubated overnight at 4 °C with a rabbit anti-c-fos primary antibody (1:5000 dilution, Synaptic Systems, Goettingen, Germany) and then with a secondary antibody (peroxidase conjugated donkey-anti-rabbit 1:500, Jackson Immunoresearch, Baltimore, PA, USA) for 1 h. For the final tyramide signal amplification and labeling step, sections from the hM4Di group were incubated for 30 min in NHS-fluorescein (1:500, Thermo Fisher Scientific, MA) sections from the eGFP group for 30 min in NHS-rhodamine (1:500, Thermo Fisher Scientific, MA) diluted in 0.1 M borate buffer with 0.01% hydrogen peroxide. Sections were washed in PBS before and after all procedures (5 × 5 min), which took place on a shaker and at room temperature. Following staining, brain slices were mounted on gelatin-coated slides and air dried before being coverslipped with Fluoroshield Mounting medium with DAPI for nuclear staining (Abcam, MA).

**Cell imaging and counting**. hM4Di and GFP expression in coronal brain slices containing the vHPC and LS were visualized at ×4 magnification using the NIKON Ni-U upright fluorescent microscope (NIKON, NY). GFP expressing cells were visualized using the FITC filter (excitation: 467–498 nm; emission: 513–556 nm), while hM4Di expressing cells were visualized using the TexRed filter (excitation: 532–587 nm; emission: 608–683 nm). Quantification of c-fos positive cells was achieved ×10 magnification stitched images of the entire LS. Both LScd and LSr cannulated animals were used to ensure sufficient power, with an ROI focused on a 0.5 mm radius around the tip of the cannula tract, with areas of damage excluded to ensure autofluorescence did not contribute toward detected signal. Additionally, areas within that radius that did not receive HPC afferents were also excluded to ensure that c-Fos positive cells that are not responsive to CNO mediated afferent inhibition did not contribute to the cell count. The number of c-Fos-positive cells within those boundaries were counted by converting the images into 8-bit and using ImageJ software (Rasband, W.S., U.S. National Institutes of Health), averaged per animal, and divided by the total area of the ROI to achieve a measurement of the number of c-fos positive cells per mm$^2$.

**Statistical analyses**. All data were analyzed using R[66] using the "tidyverse", "MKinfer", and "permuco" packages[67–69], and graphed in GraphPad Prism version 8.1.1. (GraphPad Software, La Jolla, CA). p values were derived from permutation tests with 10,000 repetitions, or by computing p values on all possible combinations where datasets had fewer than 10,000 combinations. Datasets were permuted using the method proposed by Kherad-Pajouh and Renaud[70] for mixed ANOVAs, and shuffling data labels for paired and unpaired t-tests. Results are reported with degrees of freedom, parametric test statistic, and permutated p values. Due to the number of repetitions chosen, the minimum possible p value reported is 0.0001. Significant effects and interactions were followed up with main effect analyses and pairwise comparisons, using the Bonferroni-Holm correction for multiple comparisons with adjusted permuted p-values reported. The alpha level was set at p < 0.05.

All mixed valence Y-maze acquisition and tests (mixed valence conflict test, preference tests) were subjected to ANOVA with virus as a between-subject factor and arm as a within-subject factor. The primary measure was the amount of time spent in each arm. To control for overall activity levels during the conflict test, approach-avoidance ratios were calculated, defined as Valenced Arm ÷ (Valenced Arm + Neutral Arm), where a ratio of ~0.5 indicated balanced approach and avoidance of the mixed valenced arm, and was analyzed with an unpaired t-test between groups. Secondary measures included the number of entries made into each arm, latency to enter the arm, retreats from the arm, and arm-stays. Retreats were expressed as stretches into an arm that do not result in an entry, and back-treading behavior within an arm. An arm-stay event occurred when an animal heading toward an arm exit turned around toward the well. If animals did not make any entry into a particular arm, their latency for that arm was entered as 300 s (the overall length of the test).

The novelty detection and EPM tests used virus as a between subject factor, with arm (novel vs. familiar and open vs. closed, respectively) as a within subject variable and time spent in arm was the primary measure of interest. For novel and familiar environment feeding tests, dependent variables were compared between

virus groups with unpaired *t*-tests. The primary measures included latency to make an initial bite on the food pellet, latency to begin feeding on the pellet, and the difference between these times. The familiar environment feeding test had the same dependent variables and was analyzed in the same way, with food consumed in grams as an additional measure of interest. The free feeding test was analyzed with virus as a between subject factor, and drug infusion condition and time as within subject factors. The main dependent variable was the cumulative amount of food consumed by a time point, with a supplementary analysis of food consumed within each of the 4 time bins. Locomotor behavior was analyzed with virus as a between group factor and 5 min time bin as a within-subject variable. Finally, c-Fos+ cell counts were analyzed with a three-way ANOVA with drug condition, virus, and targeted LS region as between subject factors.

**Reproducibility**. The animals in the reported experiments were tested successively in six cohorts of 6–16, with each cohort serving as mini-replicates of the entire experiment (with $n = 3$ or 4 per experimental/control group). We observed similar patterns of findings across these cohorts.

**Reporting summary**. Further information on research design is available in the Nature Research Reporting Summary linked to this article.

## Data availability
All data generated in this study have been deposited in Open Science Framework database under the accession code: https://osf.io/vs6xe/. Source data are provided with this paper.

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

## Acknowledgements

This work was supported by the Canadian Institutes of Health Research (156070) awarded to R.I. and A.C.H.L., and an Ontario Graduate Scholarship awarded to D.C.M.Y. We would like to thank Dr Maithe Arruda-Carvalho for allowing us to use and acquire images from the NIKON Ni-U upright fluorescent microscope. We also thank Dr Sandeep Dhawan, Tanner McNamara, and Norman Stewart for their help in developing immunohistochemistry protocols and/or behavioral testing.

## Author contributions

Conceptualization: A.C.H.L. and R.I.; Methodology: D.C.M.Y. and R.I.; Investigation: D.C.M.Y., D.L., S.S., N.K., and D.A. Formal analysis: D.C.M.Y.; Visualization: D.C.M.Y. and R.I.; Writing—original draft, review and editing: D.C.M.Y., A.C.H.L., and R.I.; Funding acquisition: A.C.H.L. and R.I.; Supervision: A.C.H.L. and R.I.

## Competing interests

The authors declare no competing interests.
