## [Peer Review File · Nature Communications]

Parallel Ventral Hippocampus-Lateral Septum Pathways Differentially Regulate Approach-Avoidance ConflictREVIEWER COMMENTS

Reviewer #1 (Remarks to the Author):

This paper examines the role of projections from the ventral CA3 and CA1 areas of the hippocampus to different parts of the lateral septum (LS) in the regulation of avoidance-approach. The authors used state-of-the-art chemogenetic projection-specific inhibition in combination with behavioral tests of approach-avoidance, cue preference, feeding, exploration, and innate anxiety, as well as anatomical projection tracing. They found a strong facilitation of engagement in a conflict-related behavior by inhibiting vCA3 -> LS projections, as well as a non-specific potentiation of approach by inhibiting vCA1 -> LS projections. In the absence of conflict in the same paradigm, the vCA3 manipulation slightly reduced attendance of the aversive cue. Attendance to food, which was involved in the approach-avoidance test, was also increased in another conflict task of novelty-evoked suppression of feeding. Chemogenetic manipulations had no effect on behavior in the novelty preference, innate anxiety, and locomotor activity tests. Anterograde tracing of LS efferents, according to the inputs from vCA3 and vCA1, confirmed their distinct target regions in the hypothalamus. This interesting, well-designed and thoroughly performed study sheds new light on the functions of the largely unexplored main hippocampal output to subcortical circuits via LS, indicating the differential role of ventral hippocampus to LS projections in behavioral responses to sets of cues with mixed emotional valence. While the reported effects are robust, their proper interpretation requires a more detailed presentation of the chemogenetic experimental preparation and a more comprehensive description of several findings.

Major comments:

Figure 1. "Diagram of hM4Di infection in vCA3 and cannulation in LS_{cd}, with representative DREADDs infection (red) in the vCA3 and cannula placement in the LS_{cd}." : what kind of dye was used to reveal the location of cannulas? Because the dye's colour is similar to that of mCherry and the injection site is very bright (assuming it is a dye and not mCherry-fluorescence), it is difficult to see the localisation of mCherry in parts of LS in the images provided. It would be helpful if the authors provided images of mCherry localisation without dye injections or magnified images of hippocampal projections (or lack thereof) in LS_{cd} and LS_{rv}. To better see the LS, Figs 1g and 1k could benefit from showing a smaller portion of the brain. Projections from vCA1 and vCA3 to various parts of LS are a bit difficult to see also due to bright offsetting brightness red spots on the image's periphery.

Considering the study's focus on the difference between vCA1 and vCA3 projections, and the tendency of AAVs to spread from the injection site, showing the pattern of DREADDs expression in more animals would strengthen the point of effects specificity.

P.7 “Collectively, the evidence suggests that neither vCA3->LScd nor vCA1->LSrv circuit inhibition alone has a significant effect on behaviour in response to stimuli with a single associated valence, and neither pathway affects memory or cue preferences under normal conditions.”: in Figure 5, inhibition of vCA3 projections tends to increase appetitive cue preference. The presentation of these differences, which are small when compared to the strong effects of inhibition in the conflict test, is not clear enough. Figure 5d, g: please provide statistics for the Arm x Virus interaction. Often, only main effects are provided, rather than interactions. Figure 5i: why are significant differences between the hM4Di and GFP groups reported in the legend but not shown on the figure? Figure 6 k “Both vCA3->LScd groups consumed similar amounts of food in the home cage feeding test (Time Bin: $F(3,42) = 90.29$, $P = 0.0001$).”: the statement and the statistics seem to be in conflict. Does not it show a significant affect in some time bins? Would it not make more sense to test only final intake after 4 h, and, separately, the intake in each bin, since measurements of the cumulative intake in consequent bins are related?

The visualisation of LS projections to the hypothalamus in Figure 2 is somewhat distracting from the functional study of LS afferents in this paper. The results in Figure 2 do not seem central for this paper, they are in line but do not really extend previous studies which used more specialised anatomical approaches. If the authors include anatomical results, more images taken at different rostrocaudal levels in the hypothalamus in additional animals and high-resolution images showing a counterstaining of GFP with postsynaptic markers would strengthen the message of this figure. Having said that, the references to anatomical studies already provided in the manuscript appear sufficient for the current findings' implications for further information processing in the hypothalamus.

Minor comments

Abstract, P 1 “These findings suggest that the vHPC influences multiple behavioural systems via differential downstream LS targets”: while this perspective is plausible, the current results extensively address the effects of inputs upstream of LS. Perhaps it would be more appropriate to emphasise the significance of these findings rather than of downstream LS targets.

P.3 “A complete delineation of the neural subsystems underlying such approach-avoidance conflicts”: the importance of approach-avoidance circuits and the general significance of this study can be further highlighted by referring to other regions whose role in the inhibition of conflicting responses has been extensively studied (including the subthalamic nucleus and prefrontal areas).

P.4 “Collectively, we have identified a vCA3 ->LScd pathway that suppresses exploratory responses in favour of avoidance responses,”: it seems justified to indicate that the current findings address specifically food-related exploration.

P. 13 „AAV8-CAMKII-hM4Di-mCherry or AAV8-CAMKII-EGFP control viruses (Addgene, MA) into either the ventral CA3 ...”: how were the injections done and what volumes were injected?

p.14 “have at least 4 weeks to begin expressing before attempted activation”: 4 weeks is sufficient for somatic expression, but axonal manipulations usually start about 6 weeks post-injection (also in this study, 8 weeks in anterograde tracing experiments). Was there any relationship between the time after viral infusion and behavioral effects?

P.15 “the most preferred cue was designated the aversive cue, the least explored the appetitive cue, and the remaining one the neutral cue. The third habituation session consisted of a two arm maze configuration, with one arm containing a combination of one of the “appetitive” cues and one of the “aversive” cues placed on opposite walls.”: the first sentence states that there was one cue of each type, yet then it sounds like they were several of them.

P17 “Animals were food restricted for at least 18 hrs before the free feeding task. 1 hr prior to testing (~1000-1100hrs) animals were separated from their cage mates into individual clean cages.” - because the animals were apparently housed in groups of two during food restriction (was food taken away or provided in a reduced amount?), could some rats be more stressed during this time, as rats can become aggressive to each other in these conditions? Time is misrepresented.

Each rat was subjected to a battery of tests; how many days were allowed between them, and in what order were they carried out?

If the locomotor activity test was conducted in novel cages, the results could be related to the Novelty Detection Test and would need to be distinguished from locomotor activity testing in familiar (home) cages.

P.20 “The primary measures included latency to make an initial bite on the food pellet, latency to begin feeding on the pellet, and the difference between these times”: the actual distinction between the initial bite on the food pellet and the start of feeding on the pellet, which is separated on average by 50 seconds, and the interpretation of these relevant but possibly unfamiliar to some readers measures are provided first in the discussion. It would be helpful to clarify them earlier.

P.8. “Immediately following the NESF, rats were placed within their familiar cages with 20g of pellets and allowed to consume them freely for 10 min”: the inhibition of vCA3 projections increased food

consumption in the first test; could the lack of effect in the subsequent test be due to a reduced appetite?

Which software was used to analyse locomotor activity?

Figure 1h. Why was saline used as the control in this experiment instead of CNO in GFP mice, as in other experiments presented?

Figure 5 legend: what does “post hoc virus conflict ” refer to?

Figure 6. i-j,l-m “In familiar environments all groups had similar latencies to bite (vCA3: $t_{12} = 0.62$, $P = 0.6845$;”: what comparisons are these statistics referring to? What was the statistical significance of the difference in feeding time between the GFP and hm4Di groups as shown in the figure?

P. 8 “In contrast, inactivation of the vCA1 -> LSrv circuit led to a longer latency to bite the food pellet (Figure 6e) as well as begin feeding compared to CNO infused control animals (Figure 6f).”: what are the authors' interpretations of this effect and its possible links to the conflict paradigm findings?

P11. “disrupting non-specific behavioural disinhibition”: behavioural inhibition instead?

Words marked * * seem suboptimal:

P10 “suggest that *that* neurons”, “the *valance* of negative”, “for inhibiting *select* behavioural control systems”

P. 3 “although an *adequate* and specific functional description for the region has been elusive”

P 14. to obstruct them *outside* of infusions, a

P. 4 “expression along the *body* of the vCA3”

Fig 5 legends: “...and *emitted* arm stays”

“Viral infection”: perhaps viral infusion, transduction or alike, as recombinant viruses are incapable of replicating and establishing an infection.

Reviewer #2 (Remarks to the Author):

In this elegant paper by Yeates and colleagues, the authors investigate the role of projections from the ventral hippocampus CA3 and CA1 subregions to the lateral septum (LS) during approach-avoidance conflict resolution. The authors find that chemogenetic inhibition of vCA3→LS caudodorsal (cd) projections suppresses conflict avoidance following learning of a mixed-valence approach avoidance conflict task, while inhibition of vCA1→LS rostroventral (rv) projections releases general behavioral inhibition in response to both motivationally conflicting and neutral stimuli. Important control experiments were performed to control for effects of feeding, anxiety-like behavior, locomotion, and novelty preference. The data are rigorously analyzed with appropriate statistics, and the paper is very clear and well written. The results are novel and highly relevant for our understanding of the neural circuitry underlying approach avoidance conflict behavior, which is relevant to a variety of psychiatric and neurological disorders. I only have few comments to make.

1. The NESF findings on vCA3→LS_{cd} inhibition very nicely match the findings in the conflict task, by showing that vCA3→LS_{cd} inhibition reduces conflict avoidance by shortening the latency to feed in the novel arena. However, inhibition of vCA1→LS_{rv} projections increases the latency to feed in the NESF task, while releasing general inhibition in the conflict task. How do the authors reconcile these seemingly conflicting findings? The authors argue that increased latency to feed in the NESF task may be due to disinhibited exploration of the food pellet in the novel environment, but considering that the animals were food deprived, it is difficult to imagine why increased exploration of the food pellet would not result in shorter feeding latencies. Is any data available to support this claim? It is interesting that vCA1→LS_{rv} inhibition does slightly reduce feeding in the home cage (albeit not significantly). Alternatively, is it possible that in light of these NESF findings, the data obtained from the conflict task may need to be interpreted differently?
2. For the anterograde tracing experiments in Figures 1 and 2, why were two different AAV serotypes used (8 and 1) to visualize projections from vHPC to LC vs projections from LC to downstream targets? Why were these two serotypes used instead of AAV5, which shows more efficient anterograde transport than AAV1 and AAV8?
3. The LS downstream projection targets shown in Figure 2 are highly interesting from an anatomic perspective, and provide potential hypotheses for future studies to further investigate the complex neural circuits of approach-avoidance conflict resolution. However, considering that Figures 3-7 all investigate the functional role of vHPC→LS projections, I would suggest moving the current Figure 2 to the end of the manuscript as a future direction.
4. Figure 6 panels j, m – the colors of the GFP and hM4Di bars are inverted.

Reviewer #3 (Remarks to the Author):

In the present study, the authors show evidence that hippocampal projections to the lateral septum differentially regulate approach-avoidance behaviors. Specifically, the authors suggest that hippocampal projections coming from vCA3 to LScd (caudodorsal) signal avoidance under conflict, whereas vCA1 projections to LSrv (rostroventral) promote general avoidance responses under conflict and novel scenarios (normal function without manipulations). However, while the authors used established behavioral and chemogenetic approaches that provide seminal information about approach-avoidance decisions, there are elemental limitations on innovation, organization and rationales/interpretations that detract the impact of the study to be considered for publication in Nature Communications.

Major

1. In Figure 1;

a. The authors report that while vCA3 targets the LScd, vCA1 targets LSrv, however, vCA1 projections on LS appears to reach dorsal and ventral parts (Fig 1k).

b. While the authors mention a significant decrease in cFos activity after CNO exposure, this was not necessarily supported by the present data/images (Fig 1 i-h) and/or by the statistical analyses. Providing posthoc analyses and increasing the sample size (current sample size is n=3) would help with the interpretations.

c. Providing the AP levels in the images would also help to distinguish these potential anatomical differences.

2. The addition of Figure 2 seems to be way-off and breaks the sequence of the manuscript, which is focused on hippocampal-LS pathways. This figure might be considered for supplementary information.

3. A whole composite figure to show that the animals learn a validated task is unnecessary (Fig 3). Most of this information can be placed as text (1-2 sentences) and/or in Fig 4 or in the supplementary section.

4. In figure 4, the authors suggest divergent effects on these HPC-LS projections, where inhibition of vCA3-LScd increases approach and inhibition of vCA1-LSrv promotes non-specific approach (i.e. exploration in conflict/novel scenarios). However, inhibiting the vCA1-LSrv projection showed a clear no effect during conflict. Thus, the two paragraphs suggesting trending effects on this manipulation are irrelevant (page 6). Alternatively, the authors might also consider that staying away (in the neutral arm) of the conflict arm, can be measured as active avoidance (given that the hub area may serves as the actual neutral zone).

5. Related to point 4: While inhibiting vCA3-LScd projections lead to increased approach to an arm signaling both reward and aversive cues, the behavioral task is limited in distinguishing if the lack of avoidance is the increase of approach or vice versa. Interpreting the results taking this in consideration and its further discussion would be helpful. Measuring the freezing levels or time in corners during the session might add more information about this dichotomy.
6. The lack of effects in Figure 5, can be reported in few sentences and/or in the supplementary section.
7. The other behavioral readouts depicted in Figure 7 can be described in few sentences and/or in the supplementary section. Also, while the authors reported a non-significant effect during the EPM (lack of anxiety effects), is not clear whether there is difference in the time spent in the closed arms between GFP and hM4Di groups.
8. The discussion should be more focused and precise.

Minor

1. Add reference: Page 3, Paragraph 3, First sentence.
2. Figure 1, why the authors used a CAMKII-EGFP (instead of a CAMKII-mCherry) as a control for CAMKII-hM4Di-mCherry.
3. In Figure 3, add more information/labels to help the reader. For instance, the days, the time between stages, duration of the sessions.
4. Page 7 (first two paragraphs): Should say "(Figure 5 g-h instead of e-f)" and "(Figure 5 i-n instead of g-h)".

Response Statement

We thank all the reviewers for their careful reading of our paper, and detailed constructive comments. We have made substantial revisions to our paper in response, including the addition of new animals to our dataset, primarily in response to Reviewer 3's request for increased sample size in the cFos analysis. Thus, 3 animals were added to the vCA3-hM4Di-saline group, 1 animal to the vCA3-hM4Di-CNO group, and 2 animals to the vCA1-hM4Di-saline group. In addition to this, EGFP animals were added to the cFos data, which also led to the addition of 1 animal to the vCA3-EGFP-saline group and 1 animal to the vCA1-EGFP-saline group. All newly included animals were trained and tested on the Y-maze conflict and preference tasks as well as the locomotor test and EPM. Unfortunately, due to time constraints and continued COVID-related disruption to research activities, we were not able to run these additional animals on the free feeding, NESF, or novelty tests. Nevertheless, the inclusion of additional data has strengthened our original findings. The overall pattern of behavioural findings has remained consistent and cFos findings are as predicted (i.e. a significant decline in cFos activity following CNO infusion into hM4Di animals, but not following saline infusion or CNO infusion into EGFP animals). Where relevant, statistical outputs and textual references to the results have been updated. In addition, we have clarified the text in several areas of the manuscript, edited a number of the original figures, and refined the Discussion in response to the reviewers' comments. All significant textual changes/additions in the manuscript have been highlighted. We believe that our paper is significantly improved as a result.

Reviewer 1

Reviewer #1 (Remarks to the Author):

This paper examines the role of projections from the ventral CA3 and CA1 areas of the hippocampus to different parts of the lateral septum (LS) in the regulation of avoidance-approach. The authors used state-of-the-art chemogenetic projection-specific inhibition in combination with behavioral tests of approach-avoidance, cue preference, feeding, exploration, and innate anxiety, as well as anatomical projection tracing. They found a strong facilitation of engagement in a conflict-related behavior by inhibiting vCA3 -> LS projections, as well as a non-specific potentiation of approach by inhibiting vCA1 -> LS projections. In the absence of conflict in the same paradigm, the vCA3 manipulation slightly reduced attendance of the aversive cue. Attendance to food, which was involved in the approach-avoidance test, was also increased in another conflict task of novelty-evoked suppression of feeding. Chemogenetic manipulations had no effect on behavior in the novelty preference, innate anxiety, and locomotor activity tests. Anterograde tracing of LS efferents, according to the inputs from vCA3 and vCA1, confirmed their distinct target regions in the hypothalamus. This interesting, well-designed and thoroughly performed study sheds new light on the functions of the largely unexplored main hippocampal output to subcortical circuits via LS, indicating the differential role of ventral hippocampus to LS projections in behavioral responses to sets of cues with mixed emotional valence. While the reported effects are robust, their proper interpretation requires a more detailed presentation of the chemogenetic experimental preparation and a more comprehensive description of several findings.

We thank the reviewer for their positive appraisal of our reported effects. We hope that we have been able to fully address the reviewer's concerns with the added clarity to our interpretation and experimental details.

Major comments:

Figure 1. "Diagram of hM4Di infection in vCA3 and cannulation in LScd, with representative DREADDs infection (red) in the vCA3 and cannula placement in the LScd." : what kind of dye was used to reveal the location of cannulas? Because the dye's colour is similar to that of mCherry and the injection site is very bright (assuming it is a dye and not mCherry-fluorescence), it is difficult to see the localisation of mCherry in parts of LS in the images provided. It would be helpful if the authors provided images of mCherry localisation without dye injections or magnified images of hippocampal projections (or lack thereof) in LScd and LSrv. To better see the LS, Figs 1g and 1k could benefit from showing a smaller portion of the brain. Projections from vCA1 and vCA3 to various parts of LS are a bit difficult to see also due to bright offsetting brightness red spots on the image's periphery.

We thank the reviewer for these points. We would like to qualify that what appears to be the injections of a 'dye' is in fact autofluorescence in the cannula tract (likely due to tissue damage). According to the reviewer's suggestions, we have replaced our original images with higher magnification images with less tract damage.

Considering the study's focus on the difference between vCA1 and vCA3 projections, and the tendency of AAVs to spread from the injection site, showing the pattern of DREADDs expression in more animals would strengthen the point of effects specificity.

We have implemented this in our revised Figure 2 (previously Fig 1). Fig 2e and Fig 2j depict the maximal and minimal spread of DREADDs in the vCA3 and vCA1, respectively. Additionally, since our manipulation relies on the placement of the cannulae to isolate the relevant projections from the desired HPC subfield, we have included cannula placement maps for hM4Di animals in the vCA3 to LScd group (Fig 2c) and the vCA1 to LSrv group (Fig 2h).

P.7 "Collectively, the evidence suggests that neither vCA3->LScd nor vCA1-> LSrv circuit inhibition alone has a significant effect on behaviour in response to stimuli with a single associated valence, and neither pathway affects memory or cue preferences under normal conditions.": in Figure 5, inhibition of vCA3 projections tends to increase appetitive cue preference. The presentation of these differences, which are small when compared to the strong effects of inhibition in the conflict test, is not clear enough.

We are uncertain what the reviewer is referring to here, since the time spent in the appetitive arm was comparable between hM4Di and EGFP CA3 animals. However, if the reviewer is referring to the vCA1 appetitive cue preference results, where vCA1 projection inhibited animals appear to have a higher appetitive time than the GFP controls (new Fig 4d [previously 5d]), this difference was not significant ($p = 0.12$), in keeping with our conclusion that circuit inhibition impacted behavior specifically in the conflict test.

Figure 5d,g: please provide statistics for the Arm x Virus interaction. Often, only main effects are provided, rather than interactions.

We had omitted mention of most non-significant effects due to space constraints in the figure captions, with some exceptions where non-significant results were directly pertinent to the discussion, or if there was a weak trend of interest. The Arm x Virus interactions highlighted by the reviewer for Fig 4d and g are both non-significant, at $p = 0.37$ and $p = 0.32$ respectively. We have

now included these values in the revised Figure legend for the new Fig 4d and g (previously Figure 5d, g).

Figure 5i: why are significant differences between the hM4Di and GFP groups reported in the legend but not shown on the figure?

We agree with the reviewer that these significant differences should have been indicated in the figure. In this particular case, since the effect reported was an interaction that could not be highlighted easily in a visually simple manner (i.e., without cluttering the figure), we omitted the visual reference to the significance while reporting a textual one. With the addition of further animals to the experiment, however, the interaction initially reported in Fig 4i (previously 5i) is no longer significant, and hence we have removed textual reference to it altogether.

Figure 6k "Both vCA₃-> LScd groups consumed similar amounts of food in the home cage feeding test (Time Bin: $F_{(3,42)} = 90.29$, $P = 0.0001$).": the statement and the statistics seem to be in conflict. Does not it show a significant affect in some time bins?

Thank you for pointing out the issue with this phrasing. The point was meant to demonstrate that animals, regardless of condition, consumed more food as time progresses, which was the only significant effect. Within each bin, there were no significant effects even prior to correction for multiple comparisons. We have rephrased the sentence in Fig 5k (previously 6k) to 'Both vCA₃ → LScd groups consumed more food as time progressed in the home cage feeding test (Time Bin: $F_{(3,42)} = 90.29$, $P = 0.0001$)' for increased clarity.

Would it not make more sense to test only final intake after 4 h, and, separately, the intake in each bin, since measurements of the cumulative intake in consequent bins are related?

We thank the reviewer for this comment. We initially plotted the food consumption on a bin-by-bin basis, but found that the results were visually confusing. Furthermore, the bins were not of equal duration (i.e., measurements taken at 30min, 1hr, 2hr, 4hr) and hence we felt that the data were better captured using a cumulative output, in a manner similar to that reported in Sweeney and Yang (2015, Pubmed ID: 26666960). *At the request of the reviewer, however, we have now included these graphs in the Supplementary section (Fig S2).*

When we plot the food intake in bins, we observe more variability between groups, which we believe to be random effects that are not meaningful. In both vCA₃ and vCA₁ groups, we have a significant effect of 'bin', as expected, with consumption being highest in the first 30min, and declining in the next 2 hrs, before rising again (Lowest $P = 1.34 \times 10^{-10}$).

Within the vCA₃-LScd dataset there was also a significant Bin x Drug interaction. However, this interaction appears to be driven by a higher-than-average food consumption by the hM4Di-CNO group in the 30min to 1hr bin along with lower than average food consumption by the EGFP-CNO group in the 2-4hr bin. There was no post-hoc comparison that was more significant than a trend, however, which suggests that these are very weak trends that reflect noise within the data.

Within the vCA₁-LSrv dataset there was also a Virus x Bin interaction. Again this interaction appears to be driven by noise rather than as a consistent effect of any manipulation, as the EGFP group in both drug conditions consumed less food in the 1 hr time bin than the hM4Di group.

Interestingly, the effect of drug noted in the main cumulative analysis was not present when data was analysed by bin. In both cases, there was no interaction of drug and virus, as would be expected if inhibition of either pathway played a significant role in free feeding behaviour.

Finally, when only the cumulative amount of food eaten over 4 hours is analysed, there were no significant effects in either vCA3 or vCA1 groups (lowest $p = 0.0883$).

The visualisation of LS projections to the hypothalamus in Figure 2 is somewhat distracting from the functional study of LS afferents in this paper. The results in Figure 2 do not seem central for this paper, they are in line but do not really extend previous studies which used more specialised anatomical approaches. If the authors include anatomical results, more images taken at different rostrocaudal levels in the hypothalamus in additional animals and high-resolution images showing a counterstaining of GFP with postsynaptic markers would strengthen the message of this figure. Having said that, the references to anatomical studies already provided in the manuscript appear sufficient for the current findings' implications for further information processing in the hypothalamus.

We agree with the reviewer that our tracing figure added little value to our overall findings over and above previous anatomical findings, and counterstaining for post-synaptic markers would be outside of the scope of this study and our expertise. We have, therefore, decided to move this figure to supplementary information (Supplementary Fig 3). At the request of the reviewer, we have also added further images at more rostrocaudal levels in the hypothalamus for a more complete picture of the projections.

Minor comments

Abstract, P 1 "These findings suggest that the vHPC influences multiple behavioural systems via differential downstream LS targets": while this perspective is plausible, the current results extensively address the effects of inputs upstream of LS. Perhaps it would be more appropriate to emphasise the significance of these findings rather than of downstream LS targets.

This sentence has been revised to 'These findings suggest that the vHPC influences multiple behavioural systems via differential projections to the LS'.

P.3 "A complete delineation of the neural subsystems underlying such approach-avoidance conflicts": the importance of approach-avoidance circuits and the general significance of this study can be further highlighted by referring to other regions whose role in the inhibition of conflicting responses has been extensively studied (including the subthalamic nucleus and prefrontal areas).

We have added a brief discussion of other major brain regions implicated in approach-avoidance conflict to the introduction, and contrast this with the specific types of conflict associated with the HPC (revised manuscript p. 3).

P.4 "Collectively, we have identified a vCA3 -> LS_{cd} pathway that suppresses exploratory responses in favour of avoidance responses,": it seems justified to indicate that the current findings address specifically food-related exploration.

The reviewer has a good point here. We have clarified this statement in the Introduction (revised manuscript p. 4) and have also added additional text to reflect this point in our Discussion (revised manuscript p. 12).

P. 13 „AAV8-CAMKII-hM4Di-mCherry or AAV8-CAMKII-EGFP control viruses (Addgene, MA) into either the ventral CA3 ...“: how were the injections done and what volumes were injected?

Thank you for pointing out that this information had been omitted. We have now included them in the relevant section of the methods (revised manuscript p.13).

p.14 "have at least 4 weeks to begin expressing before attempted activation": 4 weeks is sufficient for somatic expression, but axonal manipulations usually start about 6 weeks post-injection (also in this study, 8 weeks in anterograde tracing experiments). Was there any relationship between the time after viral infusion and behavioral effects?

We apologise as this was a typo on our part. Neural manipulations were in fact not attempted until 6 weeks post-surgery for all rats.

P.15 "the most preferred cue was designated the aversive cue, the least explored the appetitive cue, and the remaining one the neutral cue. The third habituation session consisted of a two arm maze configuration, with one arm containing a combination of one of the "appetitive" cues and one of the "aversive" cues placed on opposite walls.": the first sentence states that there was one cue of each type, yet then it sounds like they were several of them.

We apologise that this was not clear in the initial wording in this passage. We have now revised the text on p. 15-16 of the updated manuscript to emphasize that there were two of each cue type within an arm.

P17 "Animals were food restricted for at least 18 hrs before the free feeding task. 1 hr prior to testing (~1000-1100hrs) animals were separated from their cage mates into individual clean cages." - because the animals were apparently housed in groups of two during food restriction (was food taken away or provided in a reduced amount?), could some rats be more stressed during this time, as rats can become aggressive to each other in these conditions? Time is misrepresented.

To address the reviewer's queries, we have revised the text on p. 17 to make our food restriction methods clearer. During the 18 hrs before the free feeding task, food was removed from the cages. In terms of the stress experienced during the food removal, while we agree that food restriction in itself could be stressful to the animals, we would argue that these animals had previously been on at least 3 weeks of limited food access during radial maze training, and were therefore already habituated to the procedure. The experimenters did not find the animals noticeably more aggressive during the NESF or free feeding tasks. Furthermore, aggression between cage mates would not be an issue during the feeding test itself, as animals are fed in separate cages.

Each rat was subjected to a battery of tests; how many days were allowed between them, and in what order were they carried out?

We have added this important information in our methods section on p. 14 and created a new Figure 1 that illustrates the timeline of the study.

Briefly, animals underwent cue-outcome learning first, followed by a conflict test. Locomotor testing was then conducted ~ 1 hr following the conflict test (to minimize the number of IC infusions). Animals were then re-conditioned in the maze, and then underwent the preference/avoidance tests in counterbalanced order 2 days after the conflict test. Animals were then taken off food restriction for at least 5 days before the remainder of the tests. The novelty preference test occurred first, followed by the NESF and FESF tests, which all occurred within 2 hrs of receiving the infusion. Finally, animals were placed back on to free feeding for a full 24 hrs at least, and the final homecage feeding tests occurred.

If the locomotor activity test was conducted in novel cages, the results could be related to the Novelty Detection Test and would need to be distinguished from locomotor activity testing in familiar (home) cages.

Although it is possible that novelty of the test cage can affect behaviour during the locomotor test, ultimately, we did not find any effect of neural manipulation on locomotor activity, so it is not clear if there is any bearing of this potential novelty effect on our primary findings. Furthermore, the effect of novelty during this test should be minimal, as the cages we used were identical in dimension to their home cages, with no motivationally significant environmental stimuli.

P.20 "The primary measures included latency to make an initial bite on the food pellet, latency to begin feeding on the pellet, and the difference between these times": the actual distinction between the initial bite on the food pellet and the start of feeding on the pellet, which is separated on average by 50 seconds, and the interpretation of these relevant but possibly unfamiliar to some readers measures are provided first in the discussion. It would be helpful to clarify them earlier.

We thank the reviewer for pointing out that the readers will need more clarification on these matters. We have edited the text on p. 8 where we discuss the test, to consistently use 'exploratory bite' and 'initiation of feeding' to describe our main variables of interest.

P.8. "Immediately following the NESF, rats were placed within their familiar cages with 20g of pellets and allowed to consume them freely for 10 min": the inhibition of vCA3 projections increased food consumption in the first test; could the lack of effect in the subsequent test be due to a reduced appetite?

We have made textual revisions to enhance the clarity of this test (see revised manuscript p. 8). The animals were taken out of the apparatus immediately after the start of consumption, so the actual amount consumed would have been very minimal. Thus, we do not believe that this would have affected subsequent feeding in familiar cages.

Which software was used to analyse locomotor activity?

We used the Noldus Ethovision software to analyse locomotor activity. We have stated this information on p. 19 in the Methods.

Figure 1h. Why was saline used as the control in this experiment instead of CNO in GFP mice, as in other experiments presented?

We had initially believed that comparing CNO and saline effects within the same virus type would be the most pertinent comparison, since we require the use of a different fluorophore to visualise cfos positive cells in the hM4Di (tagged with mCherry) and GFP groups. However, we agree that a CNO-EGFP comparison is important, and have thus now included the cfos data for CNO and saline

EGFP animals for comparison with our hM4Di groups. Additionally, in response to one of reviewer 3's comments, we have added additional animals to the saline hM4Di groups. We believe that we present clear evidence that there is a significant reduction in cFos activity only when CNO is infused into animals carrying the hM4Di receptor in both vCA1 and vCA3 groups.

Figure 5 legend: what does "post hoc virus conflict" refer to?

We thank the reviewer for catching this. This refers to the comparison of retreats between the EGFP and hM4Di groups. The word 'conflict' has been changed to 'appetitive' in the legend of the new Figure 4 (previous Figure 5).

Figure 6 i-j,l-m "In familiar environments all groups had similar latencies to bite (vCA3: $t_{12} = 0.62$, $P = 0.6845$ ": what comparisons are these statistics referring to? What was the statistical significance of the difference in feeding time between the GFP and hM4Di groups as shown in the figure?

We have inserted the word 'virus' after the HPC subregion indicator to clarify that these are t tests between virus groups on a particular dependent variable (new Figure 5 legend, previously Figure 6).

P. 8 "In contrast, inactivation of the vCA1 -> LSrv circuit led to a longer latency to bite the food pellet (Figure 6e) as well as begin feeding compared to CNO infused control animals (Figure 6f)": what are the authors' interpretations of this effect and its possible links to the conflict paradigm findings?

We suggest that in environments with many motivationally significant stimuli, the vCA1 to LSrv circuit is primarily responsible for suppressing non-specific approach and exploration responses in favour of approach towards the most motivationally salient stimuli. Hence, in the absence of this circuit's activity, non-motivationally significant stimuli would be explored, such as the neutral arm in the radial maze conflict test and novel aspects of the environment in the NESF at the cost of pellet exploration (see revised manuscript p. 11-12). It is also possible that the vCA1 mediates conflict through an additional circuit, as direct inhibition of this region results in increased exploration of the neutral arm only, suggesting that another vCA1 projection should mediate conflict approach specifically (see revised manuscript p. 10).

P11. "disrupting non-specific behavioural disinhibition": behavioural inhibition instead?

Thank you for catching this. The word has been changed to 'inhibition'.

*Words marked * * seem suboptimal:*

*P10 "suggest that *that* neurons", "the *valance* of negative", "for inhibiting *select* behavioural control systems"*

*P. 3 "although an *adequate* and specific functional description for the region has been elusive"*

*P 14. to obstruct them *outside* of infusions, a*

*P. 4 "expression along the *body* of the vCA3"*

*Fig 4 (Previously 5) legends: "...and *emitted* arm stays"*

"Viral infection": perhaps viral infusion, transduction or alike, as recombinant viruses are incapable of replicating and establishing an infection.

We thank the reviewer for their detailed feedback. All these points have been addressed by finding a more pertinent choice of wording or omitting entirely. The only word we have chosen to retain is the choice of the word 'emit' which is well established in the animal learning literature to describe the production of individual behavioural responses.

Reviewer 2

Reviewer #2 (Remarks to the Author):

In this elegant paper by Yeates and colleagues, the authors investigate the role of projections from the ventral hippocampus CA₃ and CA₁ subregions to the lateral septum (LS) during approach-avoidance conflict resolution. The authors find that chemogenetic inhibition of vCA₃→LS caudodorsal (cd) projections suppresses conflict avoidance following learning of a mixed-valence approach avoidance conflict task, while inhibition of vCA₁→LS rostroventral (rv) projections releases general behavioral inhibition in response to both motivationally conflicting and neutral stimuli. Important control experiments were performed to control for effects of feeding, anxiety-like behavior, locomotion, and novelty preference. The data are rigorously analyzed with appropriate statistics, and the paper is very clear and well written. The results are novel and highly relevant for our understanding of the neural circuitry underlying approach avoidance conflict behavior, which is relevant to a variety of psychiatric and neurological disorders. I only have few comments to make.

We thank the reviewer for their positive assessment of our manuscript. We hope that we have been able to address the reviewer's concerns to the full in our revised version.

1. The NESF findings on vCA₃→LS_{cd} inhibition very nicely match the findings in the conflict task, by showing that vCA₃→LS_{cd} inhibition reduces conflict avoidance by shortening the latency to feed in the novel arena. However, inhibition of vCA₁→LS_{rv} projections increases the latency to feed in the NESF task, while releasing general inhibition in the conflict task. How do the authors reconcile these seemingly conflicting findings? The authors argue that increased latency to feed in the NESF task may be due to disinhibited exploration of the food pellet in the novel environment, but considering that the animals were food deprived, it is difficult to imagine why increased exploration of the food pellet would not result in shorter feeding latencies. Is any data available to support this claim? It is interesting that vCA₁→LS_{rv} inhibition does slightly reduce feeding in the home cage (albeit not significantly). Alternatively, is it possible that in light of these NESF findings, the data obtained from the conflict task may need to be interpreted differently?

We thank the reviewer for their thoughts. We believe that we failed to convey our explanation of the NESF data in the vCA₁-LS_{rv} group effectively in our original draft, which may have led to the reviewer misunderstanding our interpretation. We had intended to convey that the vCA₁-LS_{rv} circuit is important not for inhibiting exploration of the food pellet in the novel environment, but for inhibiting approach responses to off-target or motivationally *insignificant* stimuli. Given that increased anxiety is not a viable explanation for the increased latency to engage with the food pellet in the vCA₁-LS_{rv} inhibited group (in light of the results of the conflict and EPM tests), we argue that the observed finding is a reflection of animals spending more time exploring novel, but

'irrelevant' environmental stimuli. We have expanded this discussion p. 11-12 and supported our explanation with other work showing the activation of LS-projecting vCA1 cells during focused search of food in a previously rewarded location under extinction conditions.

2. *For the anterograde tracing experiments in Figures 1 and 2, why were two different AAV serotypes used (8 and 1) to visualize projections from vHPC to LC vs projections from LC to downstream targets? Why were these two serotypes used instead of AAV5, which shows more efficient anterograde transport than AAV1 and AAV8?*

The reviewer has a great point here. In our hands, we have found AAV5 to yield very limited transduction, particular in pathway studies, and hence resorted to using AAV 1 and 8, with which we have more confidence in inducing sufficient levels of transduction. We suspect that the spread and transduction of AAV of different serotypes may differ between mice and rats.

3. *The LS downstream projection targets shown in Figure 2 are highly interesting from an anatomic perspective, and provide potential hypotheses for future studies to further investigate the complex neural circuits of approach-avoidance conflict resolution. However, considering that Figures 3-7 all investigate the functional role of vHPC→LS projections, I would suggest moving the current Figure 2 to the end of the manuscript as a future direction.*

Thank you for this note. Reviewer 1 raised a similar point, and hence we have now moved this anatomical tracing figure to Supplemental information.

4. *Figure 6 panels j, m – the colors of the GFP and hM4Di bars are inverted.*

We thank the reviewer for picking this up. We have corrected this mistake in the new Figure 5 (previously Figure 6).

Reviewer 3

Reviewer #3 (Remarks to the Author):

In the present study, the authors show evidence that hippocampal projections to the lateral septum differentially regulate approach-avoidance behaviors. Specifically, the authors suggest that hippocampal projections coming from vCA3 to LScd (caudodorsal) signal avoidance under conflict, whereas vCA1 projections to LSrv (rostroventral) promote general avoidance responses under conflict and novel scenarios (normal function without manipulations). However, while the authors used established behavioral and chemogenetic approaches that provide seminal information about approach-avoidance decisions, there are elemental limitations on innovation, organization and rationales/interpretations that detract the impact of the study to be considered for publication in Nature Communications.

We thank the reviewer for their assessment of our manuscript. We believe the innovation of our paper lies in the application of pathway-specific chemogenetics to a relatively newly established custom-designed learned approach-avoidance behavioural task. This task addresses the limitations of traditional tasks that dominate the field (e.g., EPM) by capturing approach-avoidance decisions based on concrete learned contingencies, rather than innate reactions. Critically, we believe that our work sheds novel light on two neural circuits, vCA3 to LScd and vCA3 to LSrv, which have previously been hypothesized, but not yet, to our knowledge, been delineated to subserve any functional roles in the literature. We hope that the reviewer will find our revised

manuscript improved in organization and focus, and recognise the conceptual advances our findings bring in dissociating two ventral hippocampal-lateral septal pathways that are differentially involved in deploying behavioural inhibition in the face of motivational conflict.

Major

1. In Figure 1;

a. The authors report that while vCA3 targets the LScd, vCA1 targets LSrv, however, vCA1 projections on LS appears to reach dorsal and ventral parts (Fig 1k).

We would like to clarify that what appears to be viral expression in more dorsal parts of the LS are, in fact, autofluorescence we often observe where there is cannula tract damage. As requested by Reviewer 1, we have now magnified the images in the new Figure 2 (previously Figure 1) to show that the majority of the signal is in the ventral LS.

b. While the authors mention a significant decrease in cFos activity after CNO exposure, this was not necessarily supported by the present data/images (Fig 1 i-h) and/or by the statistical analyses. Providing posthoc analyses and increasing the sample size (current sample size is n=3) would help with the interpretations.

We agree with the reviewer that a sample size of 3 animals in one of our cFos groups needed to be strengthened, and hence we have increased the sample size such that we now have a minimum of n=5 in each group. The data now show very clear statistical evidence of a reduction in cFos activity selectively in the group of hM4Di-expressing animals that received CNO. Additionally, we have included the results of cFos analysis conducted in our EGFP groups to address a point raised by Reviewer 1.

c. Providing the AP levels in the images would also help to distinguish these potential anatomical differences.

We have now added the AP levels in the images.

2. The addition of Figure 2 seems to be way-off and breaks the sequence of the manuscript, which is focused on hippocampal-LS pathways. This figure might be considered for supplementary information.

Thank you for this point. The other reviewers raised the same point, and hence we have now moved this figure to supplementary information.

3. A whole composite figure to show that the animals learn a validated task is unnecessary (Fig 3). Most of this information can be placed as text (1-2 sentences) and/or in Fig 4 or in the supplementary section.

We agree with the reviewer's assessment and have moved the acquisition data figure to the supplementary section (Supplementary Figure 1). Within the main text, we have cut down the explanation of this figure, only leaving sufficient detail of our variables of interest for readers that may be unfamiliar with the task (page 5).

4. In figure 4, the authors suggest divergent effects on these HPC-LS projections, where inhibition of vCA3-LScd increases approach and inhibition of vCA1-LSrv promotes non-specific approach (i.e. exploration in

conflict/novel scenarios). However, inhibiting the vCA1-LSrv projection showed a clear no effect during conflict. Thus, the two paragraphs suggesting trending effects on this manipulation are irrelevant (page 6).

We believe that the reviewer may have misinterpreted our results here. We did observe a significant effect of chemogenetic manipulation in the vCA1-LSrv pathway on the overall amount of time spent in both the conflict and neutral arms, when compared with the EGFP control groups, and hence this was not a trend. This effect was, in fact, strengthened after the addition of more animals in the experiment, and is now supported by other measures such as the number of retreats (i.e., vCA1-LSrv inhibited animals exhibit significantly reduced retreats compared to the EGFP group).

Alternatively, the authors might also consider that staying away (in the neutral arm) of the conflict arm, can be measured as active avoidance (given that the hub area may serve as the actual neutral zone).

The reviewer offers a very interesting account of the animals entering and staying in the neutral arm. It is certainly plausible that the neutral arm could offer a 'safe' arm, akin to a one-way shuttle box paradigm. However, we do not believe this to be the best interpretation of our present data for the following reasons. Firstly, we have seen across many studies (published and unpublished) that our control animals do not preferentially stay in the neutral arm when confronted with a choice between the conflict and neutral arms. If animals were truly actively avoiding the conflict arm, we would expect to observe pronounced avoidance bias. With regards to the present findings, we found that vCA1-LSrv pathway inhibition elevated the time spent in both arms (equally) rather than just the neutral arm, and therefore sustained active avoidance is not consistently present throughout the length of the test. Additionally, the active avoidance interpretation would also be expected to be coupled with other signs of avoidance of the conflict arm, such as lower arm entries, more retreats, or a longer latency to enter. In our data there was either no change in these factors, or in the case of retreats, there was an overall reduction. Therefore, we do not think that alterations in active avoidance could explain the results of our hM4Di pathway inhibition groups.

5. Related to point 4: While inhibiting vCA3-LScd projections lead to increased approach to an arm signaling both reward and aversive cues, the behavioral task is limited in distinguishing if the lack of avoidance is the increase of approach or vice versa. Interpreting the results taking this in consideration and its further discussion would be helpful, Measuring the freezing levels or time in corners during the session might add more information about this dichotomy.

We agree with the reviewer that the conflict test in and of itself may not provide *definitive* insight into whether an animal approaches more under conflict due to an increased approach vs. decreased avoidance. Furthermore, based on the present data, we are not able to distinguish between these two possibilities. However, we have added additional discussion (on page 10) to argue that it may be more productive to interpret our data in the light of the role of the vHPC in behavioural inhibition. Thus, the potentiated approach bias we observed in the vCA3-LScd inhibition group in the face of motivational conflict may be considered to reflect disinhibited approach towards reward stimuli.

As to the reviewer's point concerning freezing levels or time in corners being additional measures that could be informative, we have reasons to believe that they would not be the most appropriate measures of approach/avoidance in our task. Firstly, due to the shock levels being kept low (0.30-

0.42mA), animals rarely freeze in this apparatus, and remain highly mobile even after exposure to the aversive cue. If there is immobility, then it is accompanied by exploratory behaviour (rearing, sniffing), making a link between freezing and aversion problematic. Secondly, the dimensions of the arms are very modest, such that there is simply no corner in which rats can cower and escape to.

6. The lack of effects in Figure 5, can be reported in few sentences and/or in the supplementary section.

We have carefully considered the reviewer's point, but have decided in favour of keeping Figure 4 (Previously 5) in the main manuscript for two reasons. Firstly, we did find a significant effect of chemogenetic inhibition on the vCA1-LSrv retreat behaviour, which provides important insight into the nature of the pattern of deficit observed in the vCA1-LSrv group. Secondly, the preference/avoidance test are extremely important in ruling out fundamental changes in appetitive and aversive motivation in accounting for any alterations seen in the conflict test. We believe that it would be of interest to the readers to see that appropriate control experiments were conducted in our study.

7. The other behavioral readouts depicted in Figure 7 can be described in few sentences and/or in the supplementary section. Also, while the authors reported a non-significant effect during the EPM (lack of anxiety effects), is not clear whether there is difference in the time spent in the closed arms between GFP and hM4Di groups.

We have followed the suggestion of the reviewer, and have moved the additional tests we conducted (Novelty preference, EPM and locomotor activity) to supplementary information (now Supplementary Figure 3).

For the EPM closed arm comparison, both vCA3 ($p = 0.32$) and vCA1 group ($p = 0.08$) comparisons were non-significant with correction.

8. The discussion should be more focused and precise.

We agree with this assessment and have made substantial changes to the discussion. We have made textual excisions and wording changes throughout to clarify some points that we felt could have been made clearer, including a restructuring of how the NESF results were discussed. Additionally, in response to one of reviewer 1's comments we have added a small section discussing the generalizability of the findings at the end of the fifth paragraph of the discussion on p. 12.

Minor

- 1. Add reference: Page 3, Paragraph 3, First sentence.*

Thank you for pointing this omission out. We have now corrected this.

- 2. Figure 1, why the authors used a CAMKII-EGFP (instead of a CAMKII-mCherry) as a control for CAMKII-hM4Di-mCherry.*

We completely agree that the best control for our CAMKII-hM4Di-mCherry would have been the use of a CAMKII-mCherry. This was purely based on the fact that at the time of our study, CAMKII-

Yeates et al.

mCherry was not commercially available (e.g., Addgene), and hence we were not able to source this AAV.

- 3. In Figure 3, add more information/labels to help the reader. For instance, the days, the time between stages, duration*

Thank you for this point. We have added the required information to the new Figure 1 (previously Figure 3).

- 4. Page 7 (first two paragraphs): Should say "(Figure 5 g-h instead of e-f)" and "(Figure 5 i-n instead of g-h)".*

Thank you for pointing out the incorrect directions in the text. We have now corrected this.

REVIEWERS' COMMENTS

Reviewer #1 (Remarks to the Author):

The authors nicely revised the manuscript, adequately addressing my concerns. This thorough study makes an important contribution to several dynamic fields, specifically behavioural functions of hippocampal subregions, hippocampal-subcortical signalling and neural circuits involved in the processing of conflicting alternatives. I have several additional minor suggestions:

Fig 5k, legend, the previously stated lack of difference between treatments, would make it more complete, i.e., "(Time Bin: $F(3,42) = 90.29$, $P = 0.0001$, Treatment: ..)'

Figure S2. It is nice to see the effects of both CNO and Saline injections together, but combining them in one ANOVA unnecessarily adds an additional factor. On the other hand, it is not stated that the ANOVA was repeated within animals, as the design seems to demand. A repeated two-way ANOVAs with factors Time bin and Virus, separately for CNO and Saline, seems more appropriate for Fig. S2.

Line 94 "Collectively, we have identified a vCA3 - LScd pathway that suppresses exploratory responses towards appetitive stimuli in favour of avoidance responses, and a vCA1- LSrv pathway that non-specifically attenuates approach responses towards motivationally salient stimuli under situations of learned and innate approach-avoidance conflict." sounds like appetitive and motivationally salient stimuli are somehow different. Please modify this as well as "in situations" instead of "under situations".

Line 363 „have observed HPC activity" should probably be "..HPC activation".

Line 397 Please specify the injection equipment used.

Fig 1. "CNO-saline" and "CNO or saline" appear side by side.

Line 320 "The behavioral phenotypes" would imply not applicable here genotypic differences, it should be e.g. "The behavioral effects".

Reviewer #2 (Remarks to the Author):

In this revision of their manuscript, the authors have been very responsive to the reviewers' concerns and added more mice and further explanations and clarifications, which have greatly improved the manuscript. I consider this paper now suitable for publication in Nature Communications.

Reviewer #3 (Remarks to the Author):

The authors have done a solids job in revising the manuscript. I have nothing further.

Response to Reviewers

Reviewer 1

We thank the reviewer again for their careful reading of our manuscript.

Fig 5k, legend, the previously stated lack of difference between treatments, would make it more complete, i.e., '(Time Bin: $F(3,42) = 90.29, P = 0.0001, Treatment: \dots$)'

This has been corrected in Figure 5k legend.

Figure S2. It is nice to see the effects of both CNO and Saline injections together, but combining them in one ANOVA unnecessarily adds an additional factor. On the other hand, it is not stated that the ANOVA was repeated within animals, as the design seems to demand. A repeated two-way ANOVAs with factors Time bin and Virus, separately for CNO and Saline, seems more appropriate for Fig. S2.

We apologise for the confusion here, but for this behavioural task only, we used a within-subject design, with each subject receiving both saline and drug CNO. Thus, the data need to be analysed together with a repeated measures 3 way ANOVA (2 within-subject factors of drug and time bin, and 1 between-subject factor of virus). This has been clarified in the figure caption (Fig S2).

Line 94 "Collectively, we have identified a vCA₃ - LS_{cd} pathway that suppresses exploratory responses towards appetitive stimuli in favour of avoidance responses, and a vCA₁- LS_{rv} pathway that non-specifically attenuates approach responses towards motivationally salient stimuli under situations of learned and innate approach-avoidance conflict." sounds like appetitive and motivationally salient stimuli are somehow different. Please modify this as well as "in situations" instead of "under situations".

We have made this suggested change.

Line 363 „have observed HPC activity" should probably be "..HPC activation".

We agree, and have edited the sentence.

Line 397 Please specify the injection equipment used.

This information has been added.

Fig 1. "CNO-saline" and "CNO or saline" appear side by side.

We could not find incidence of this typo but have carefully checked the figure and figure legend for any errors.

Line 320 "The behavioral phenotypes" would imply not applicable here genotypic differences, it should be e.g. "The behavioral effects".

Thank you for this suggestion – we agree, and have applied the change.